# Chronic activation of a negative engram induces behavioral and cellular abnormalities

**Alexandra L Jellinger**[1†], **Rebecca L Suthard**[1,2†], **Bingbing Yuan**[3], **Michelle Surets**[1], **Evan A Ruesch**[1], **Albit J Caban**[1,2], **Shawn Liu**[4], **Monika Shpokayte**[1,2]*, **Steve Ramirez**[1,5,6]*

[1]Department of Psychological and Brain Sciences, The Center for Systems Neuroscience, Boston University, Boston, United States; [2]Graduate Program for Neuroscience, Boston University, Boston, United States; [3]Whitehead Institute for Biomedical Research, MIT, Cambridge, United States; [4]Department of Physiology and Cellular Biophysics, Columbia University Medical Center, New York, United States; [5]Neurophotonics Center, and Photonics Center, Boston University, Boston, United States; [6]Department of Biomedical Engineering, Boston University, Boston, United States

*For correspondence:
mshpokay@bu.edu (MS);
dvsteve@bu.edu (SR)

†These authors contributed equally to this work

Competing interest: The authors declare that no competing interests exist.

**Abstract** Negative memories engage a brain and body-wide stress response in humans that can alter cognition and behavior. Prolonged stress responses induce maladaptive cellular, circuit, and systems-level changes that can lead to pathological brain states and corresponding disorders in which mood and memory are affected. However, it is unclear if repeated activation of cells processing negative memories induces similar phenotypes in mice. In this study, we used an activity-dependent tagging method to access neuronal ensembles and assess their molecular characteristics. Sequencing memory engrams in mice revealed that positive (male-to-female exposure) and negative (foot shock) cells upregulated genes linked to anti- and pro-inflammatory responses, respectively. To investigate the impact of persistent activation of negative engrams, we chemogenetically activated them in the ventral hippocampus over 3 months and conducted anxiety and memory-related tests. Negative engram activation increased anxiety behaviors in both 6- and 14-month-old mice, reduced spatial working memory in older mice, impaired fear extinction in younger mice, and heightened fear generalization in both age groups. Immunohistochemistry revealed changes in microglial and astrocytic structure and number in the hippocampus. In summary, repeated activation of negative memories induces lasting cellular and behavioral abnormalities in mice, offering insights into the negative effects of chronic negative thinking-like behaviors on human health.

## eLife assessment

This **useful** study reports the behavioral and physiological effects of the longitudinal activation of neurons associated with negative experiences. The main claims of the article are supported by **solid** experimental evidence, although the specificity of the long-term manipulation could have benefitted from additional validation. This study will be of interest to neuroscientists working on memory.

## Introduction

Stress has profound impacts on both physical and mental health. In particular, chronic stress is recognized as a potent risk factor for numerous neuropsychiatric disorders, including major depressive

disorder (MDD), generalized anxiety disorder (GAD), and post-traumatic stress disorder (PTSD). In humans, repeated negative thinking can parallel the effects of chronic stress. These effects include hippocampal hyperactivity and accelerating the onset of mild cognitive impairment and dementias (*Marchant et al., 2020*). Additionally, the ability of memories to influence cognition and behavior has been well established. For example, our lab previously demonstrated that reactivating ensembles of hippocampal neurons encoding a positive memory is sufficient to induce anti-depressant effects in mice (*Ramirez et al., 2015*), and studies in rodents have shown that repeated reactivation of negative memories enhances fear responses (*Chen et al., 2019*). However, it remains unknown whether modulating a chronic negative memory is sufficient to induce aberrant cellular and behavioral states that recapitulate chronic stress.

The ventral hippocampus (vHPC), specifically vCA1, has been extensively studied for its role in emotional processing. As a modulator of stress responses, vCA1 dysfunction is intimately associated with pathological affective states (Fanselow and Dong, 2010), and its structure and function are heavily impacted by stressors. Additionally, vCA1 neurons project to various downstream targets and route target-specific streams of information to the basolateral amygdala, nucleus accumbens, and prefrontal cortex. These diverse connections are integral to the formation of emotional memory, as well as regulating anxiety and the stress response. Within vCA1, stimulation of negative engram cells has been shown to modulate social avoidance behavior and enhance fear responses, but whether chronic stimulation of fearful memories produces enduring effects on brain activity and behavior remains unknown (*Chen et al., 2019*; *Zhang et al., 2019*).

Young and old individuals are differentially susceptible to the hippocampal effects of stress and chronic negative thinking behaviors. Epidemiological population-based studies in humans have established adverse childhood experiences as a risk factor for the development of psychopathologies later in life. Notably, the development of depression and PTSD may increase with the number of these adverse events, suggesting an increased susceptibility of the developing, young brain to the perturbations induced by such events (*McLaughlin et al., 2010*). However, work in older humans and rodents has shown that there are changes in their stress response and cortisol levels following adverse experiences. Aged rodents have increased stress vulnerability, as repeated stress increased anxiety-like behaviors, induced spatial memory impairments, and raised corticosterone levels more compared to younger animals (*Wang et al., 2020*).

Here, we chronically reactivated negative memory engrams to mimic the effects of stress or chronic negative thinking-related behaviors in mice. We first characterized the genetic landscape of negative and positive memory engrams to search for putative genetic markers involved in healthy and dysfunctional brain states. We then tested whether stimulating negative memory-bearing neurons would be sufficient to induce cellular abnormalities as well as behavioral impairments in young (6 months) and aged (14 months) mice. Our manipulation induced behavioral abnormalities that manifested as increased anxiety, decreased spatial working memory, and increased fear response during extinction and generalization. We also observed cellular alterations, including changes in the number and morphology of microglia and astrocytes. Together, our data suggest that negative memory engrams are a potent means by which to induce cellular and behavioral changes that provide insight into pathological states.

## Results

### Ventral hippocampal negative and positive engram cells differentially express genes associated with inflammation

The vHPC parses out negative and positive memories into anatomically segregated, functionally distinct, and molecularly diverse neuronal ensembles or 'engrams' that have unique transcriptional profiles and DNA methylation landscapes (*Shpokayte et al., 2022*). As positive and negative memories have been respectively implicated in driving approach and avoidance-related behaviors in mice, we first sought to test if these two vHPC engrams differentially upregulated genes in response to each experience. We previously performed an RNA-seq experiment described in *Shpokayte et al., 2022*. Briefly, we utilized an activity-dependent labeling technique to visualize and manipulate the cells processing a specific memory in the hippocampus by leveraging the cFos protein as an indicator of neuronal activity (*Liu et al., 2012*). Our approach combines a doxycycline (DOX)-dependent strategy

with a dual-viruses approach consisting of AAV9-c-fos-tTA+AAV9-TRE-ChR2-eYFP or control AAV9-TRE-eYFP, 24 hr prior to a behavioral experience, DOX is removed from the animal's diet which will allow tTa to bind to TRE and drive the expression of any downstream effector, such as the fluorescent protein eYFP (*Figure 1a*). After the behavioral experience, that is, contextual fear conditioning (CFC) or male-to-female exposure, the animal is placed back on DOX to close the tagging window for the negative or positive experience. Thus, the cells involved in processing a discrete experience express the eYFP protein that can be used in fluorescence-activated cell sorting (FACS) to isolate only the eYFP+ as 'tagged' engram cells for genetic sequencing. To analyze this RNA-seq dataset, we selected specific gene sets and conducted Gene Set Enrichment Analysis (GSEA) to compare the up- and downregulation of these genes in positive or negative engram cells. The GSEA showed that the anti-inflammatory gene set was enriched in positive engram cells, with higher expression compared to negative engram cells (*Figure 1c and d*). Further, the pro-inflammatory gene set was enriched in negative engram cells, with higher expression compared to positive engram cells (*Figure 1e and f*). Interestingly, this gene set included *Spp1*, *Ttr*, and *C1qb1*, which are involved in inflammatory processes and oxidative stress. Together, these data provide the start to uncovering the molecular landscape that positive and negative memory-bearing cells may contain that links them to protective or pathological functioning, respectively.

## Chronic engram stimulation increases anxiety-related behaviors, decreases spatial memory, and impairs fear extinction and generalization

Mouse models of chronic stress display increased anxiety-related behaviors and impairments in cognition and memory-related tasks that are similar to those seen in human patients with MDD, PTSD, and comorbid GAD (*Tran and Gellner, 2023*). To further understand the impact of negative engram cells, we next asked if chronic stimulation of a negative engram could induce behavioral changes in anxiety and memory-related tasks. To that end, we combined excitatory Designer Receptors Exclusively Activated by Designer Drugs (DREADDs) with a Fos-based transgenic mouse, Fos-tm2.1(iCre-ERT2) raised on a C57BL/6 background (TRAP2), to label active cells under the control of 4-hydroxytamoxifen (4-OHT) (*DeNardo et al., 2019*). Then, 3- and 11-month-old TRAP2 mice were bilaterally injected with a Cre-dependent AAV-DIO-hM3Dq-mCherry or control AAV-DIO-mCherry into the vHPC. After 2 weeks for viral expression and surgical recovery, all mice were subjected to CFC, consisting of four, 1.5 mA foot shocks in context A (Cxt A). Also, 30 min after CFC, mice received a 40 mg/kg injection of 4-OHT to label cells sufficiently active during this fearful experience (*Figure 2a and b*). The animals were returned to their home cages, where they received water-soluble deschloroclozapine dihydrochloride (DCZ), a highly potent, selective, and brain-penetrable muscarinic hM3D(Gq) actuator, for 3 months in their daily water (*Nagai et al., 2020*). DCZ was selected because even at very low concentrations this ligand is able to increase neuronal activity with binding to hM3D(Gq) receptors (*Nagai et al., 2020*). To avoid unnecessary pain or stress to the animal with chronic administration of DCZ intraperitoneally (IP), we selected the voluntary oral route (PO), which has been utilized in the past for chemogenetic experiments (*Zhan et al., 2019*) and shows no significant differences in behavior at the same dose (*Ferrari et al., 2022*).

After 3 months of negative engram stimulation, mCherry and hM3Dq mice were aged to 6 and 14 months old at the time of behavioral testing (*Figure 2a*). The weights of both the young and old mice increased independently of hM3Dq activation across the 3 months of stimulation, suggesting normal weight gain with aging (two-way repeated measures [RM] ANOVA; [6 months] interaction: $F_{(2, 34)} = 0.2633$, p=0.7701; month: $F_{(2, 34)} = 14.84$, p<0.0001; group: $F_{(1,17)} = 1.361$, p=0.2594; subject: $F_{(17,34)} = 15.44$, p<0.0001. Sidak's [month]: 1 vs. 2, p=0.0279; 1 vs. 3, p<0.0001; 2 vs. 3, p=0.0323. [14 months] Interaction: $F_{(2,52)} = 2.607$, p=0.0833; month: $F_{(2,52)} = 7.854$, p=0.0010; group: $F_{(1,26)} = 0.1275$, p=0.7239; subject: $F_{(26,52)} = 20.74$, p<0.0001. Sidak's [month]: 1 vs. 2, p=0.6942; 1 vs. 3, p=0.0011; 2 vs. 3, p=0.0198) (*Figure 2c*). However, we note that we did not include a month 0 body weight measurement; thus, it is possible that during the first month of stimulation (months 0–1), there may have been a drop in body weight that rebounded by the first measurement at month 1 that continued to increase normally through months 2–3 as shown in *Figure 1*.

Importantly, there were no decreases in the number of NeuN+ cells in the subiculum, dentate gyrus, and CA1 of the vHPC due to chronic activation, suggesting that we are not inducing a reduction

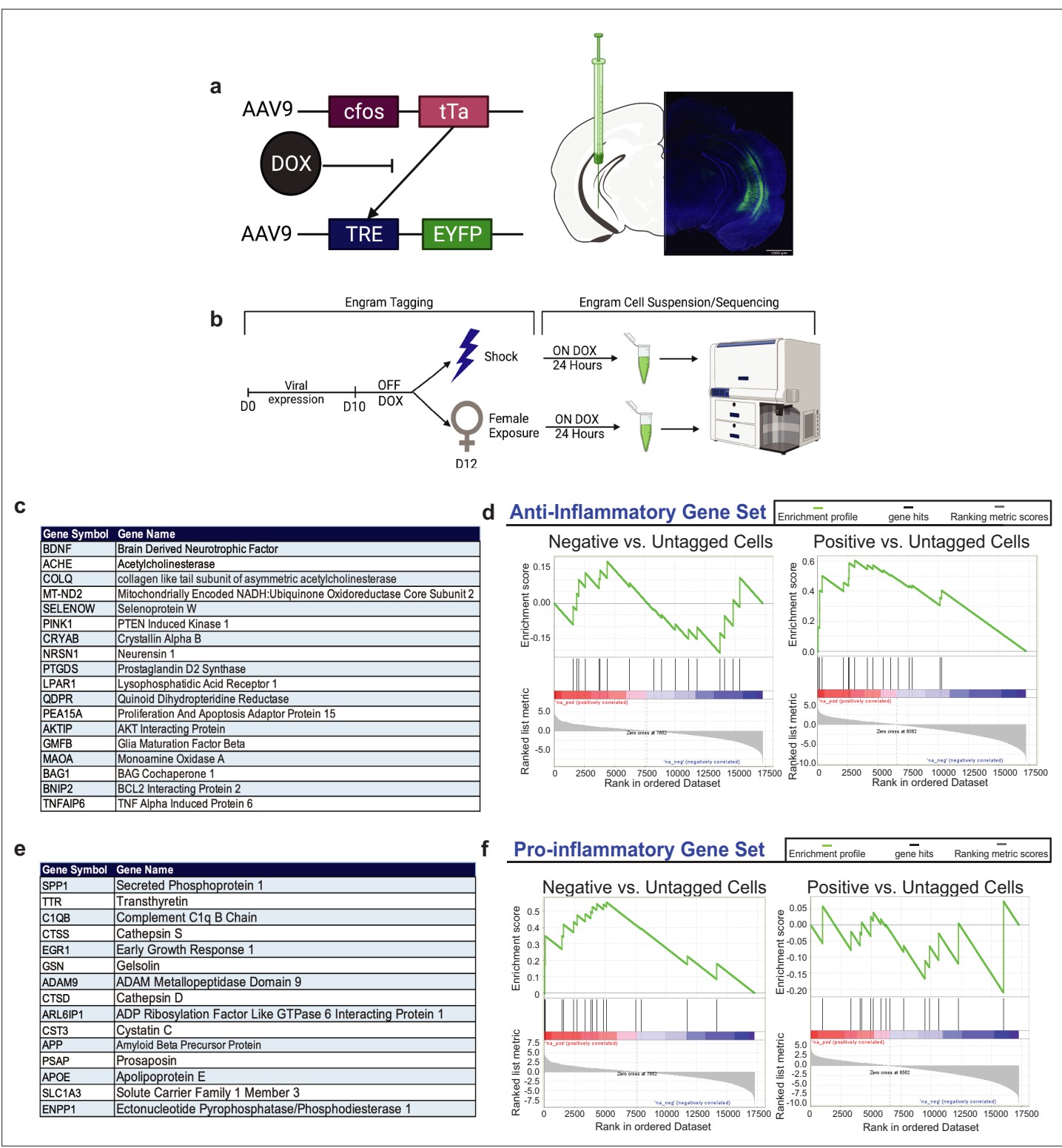

**Figure 1.** Gene Set Enrichment Analysis (GSEA) of RNA-sequencing data of positive and negative ventral hippocampus (vHPC) engram upregulate genes associated with neuroprotective and neurodegeneration, respectively. (**a**) C57BL/6J mice were bilaterally injected with a 1:1 viral cocktail of AAV9-c-Fos-tTA and AAV9-TRE-eYFP into the vHPC. This activity-dependent labeling strategy captures sufficiently active neurons expressing the immediate-early gene, *cfos*. This system couples the *cfos* promoter to the expression of the tetracycline transactivator (tTA), which binds to the tetracycline response element (TRE) in its protein form. When doxycycline (DOX) is present, such as in the animal's diet, this inhibits the binding of tTA to TRE, preventing eYFP labeling of active cells. (**b**) Experimental schematic used to label, isolate, and analyze the two groups of vHPC cells labeled by eYFP

*Figure 1 continued on next page*

*Figure 1 continued*

upon shock (negative) or male-to-female interaction (positive). Prior to surgery, mice were placed on a Dox diet to inhibit the labeling of active neurons. On day 0, mice were injected with the activity-dependent viral strategy to enable labeling of active neurons during a salient experience. On day 10 after viral expression and recovery, the mice were taken off of Dox prior to engram tagging to open the labeling window. On day 11, mice were subjected to contextual fear conditioning (CFC) or male-to-female interaction to label negative or positive neurons, respectively. Mice were placed immediately back on Dox to close this tagging window. Then, 24 hr later, mice were sacrificed and brains were obtained for sequencing experiments. (**c**) List of gene abbreviations and full names used for the GSEA of anti-inflammatory genes. (**d**) GSEA with the enrichment score for negative vs. untagged cells and positive vs. untagged cells using the anti-inflammatory gene set. (**e**) List of gene abbreviations and names used for the GSEA of pro-inflammatory genes. (**f**) GSEA with the enrichment score of negative vs. untagged cells and positive vs. untagged cells using the pro-inflammatory gene set.

The online version of this article includes the following figure supplement(s) for figure 1:

**Figure supplement 1.** Chronic stimulation of the negative engram does not induce neuronal death.

in neuronal cell health with this constant stimulation, as measured by this method (two-way ANOVA; [subiculum] interaction: $F_{(1, 8)} = 0.2516$, p=0.6295; age: $F_{(1, 8)} = 7.341$, p=0.0267; group: $F_{(1,8)} = 0.00721$, p=0.9344. Tukey's [group]: 6MO mCherry vs. hM3Dq, p=0.9904; 14MO mCherry vs. hM3Dq, p=0.9744; *Figure 1—figure supplement 1a and b*. [vDG] Interaction: $F_{(1,8)} = 0.6806$, p=0.4333; age: $F_{(1,8)} = 4.423$, p=0.0686; group: $F_{(1,8)} = 0.0524$, p=0.8246; *Figure 1—figure supplement 1c and d*.

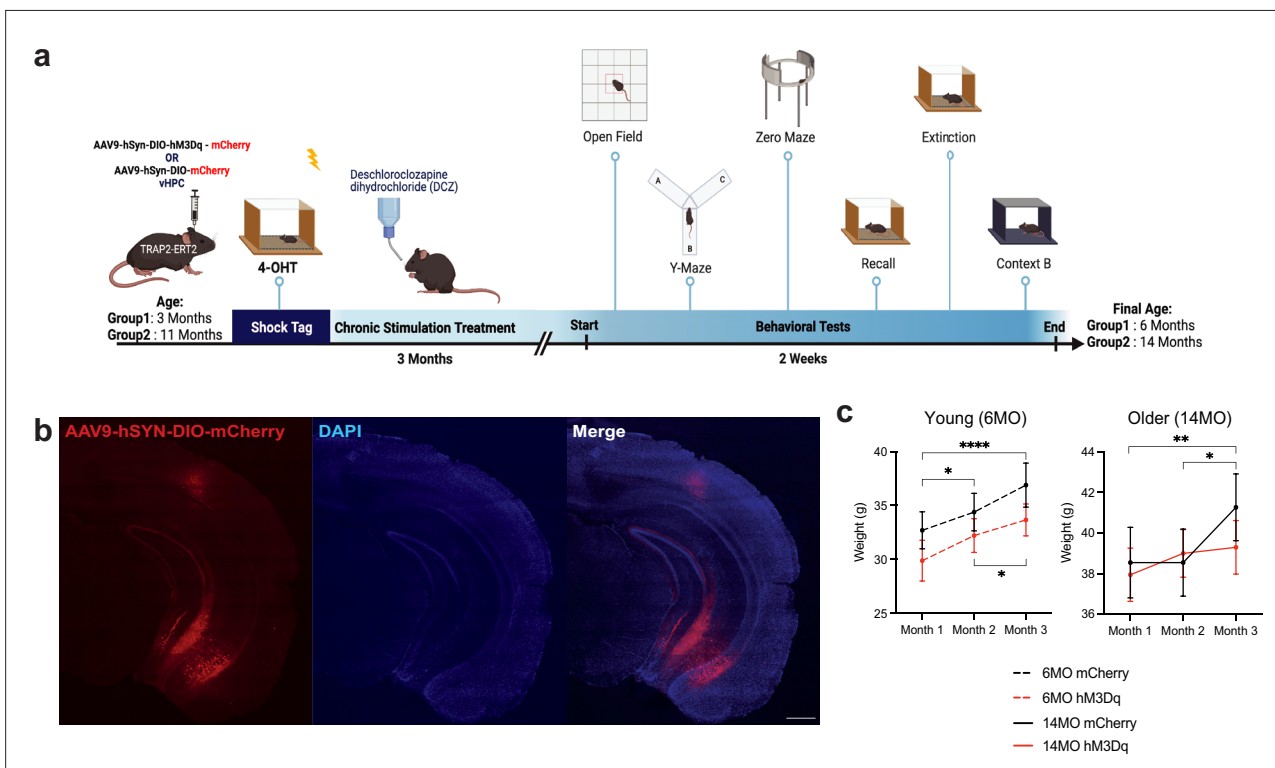

**Figure 2.** Experimental timeline for chronic modulation of a negative engram in TRAP2 mice. (**a**) Schematic representation of tagging a negative engram in young (3 months) and old (11 months) TRAP2 mice followed by 3 months of chronic engram stimulation and a battery of behavioral tests. AAV9-hSyn-DIO-hM3Dq-mCherry or AVV9-hSyn-DIO-mCherry control vector was bilaterally injected into the ventral hippocampus (vHPC) of young and old mice. After viral expression and surgical recovery, all mice were subjected to contextual fear conditioning (CFC) in context A and subsequent 4-hydroxytamoxifen (4-OHT) intraperitoneal (IP) injection to induce negative engram tagging. For the next 3 months, mice received the water-soluble Designer Receptors Exclusively Activated by Designer Drugs (DREADDs) agonist deschlorclozapine dihydrochloride (DCZ) in their home cage water. After 3 months of stimulation wherein the young and old groups reached 6 months and 14 months of age, the mice underwent open field, y-maze, zero maze, remote recall, extinction, and generalization in a novel context B. (**b**) Representative image of hSyn-DIO-hM3Dq-mCherry expression in the vHPC (red) and DAPI+ cells (blue) after 3 months of hM3Dq activation. (**c**) Weights of all groups were recorded over the 3-month stimulation protocol, once per month. (Left) Young 6-month-old and (right) older 14-month-old mice. Values are given as a mean + SEM. Statistical analysis was performed with two-way (RM) ANOVA followed by Sidak's post hoc test. n = 9–17 per group after outlier removal. p≤0.05, **p≤0.01, ***p≤0.001, ****p≤0.0001, no label = not significant.

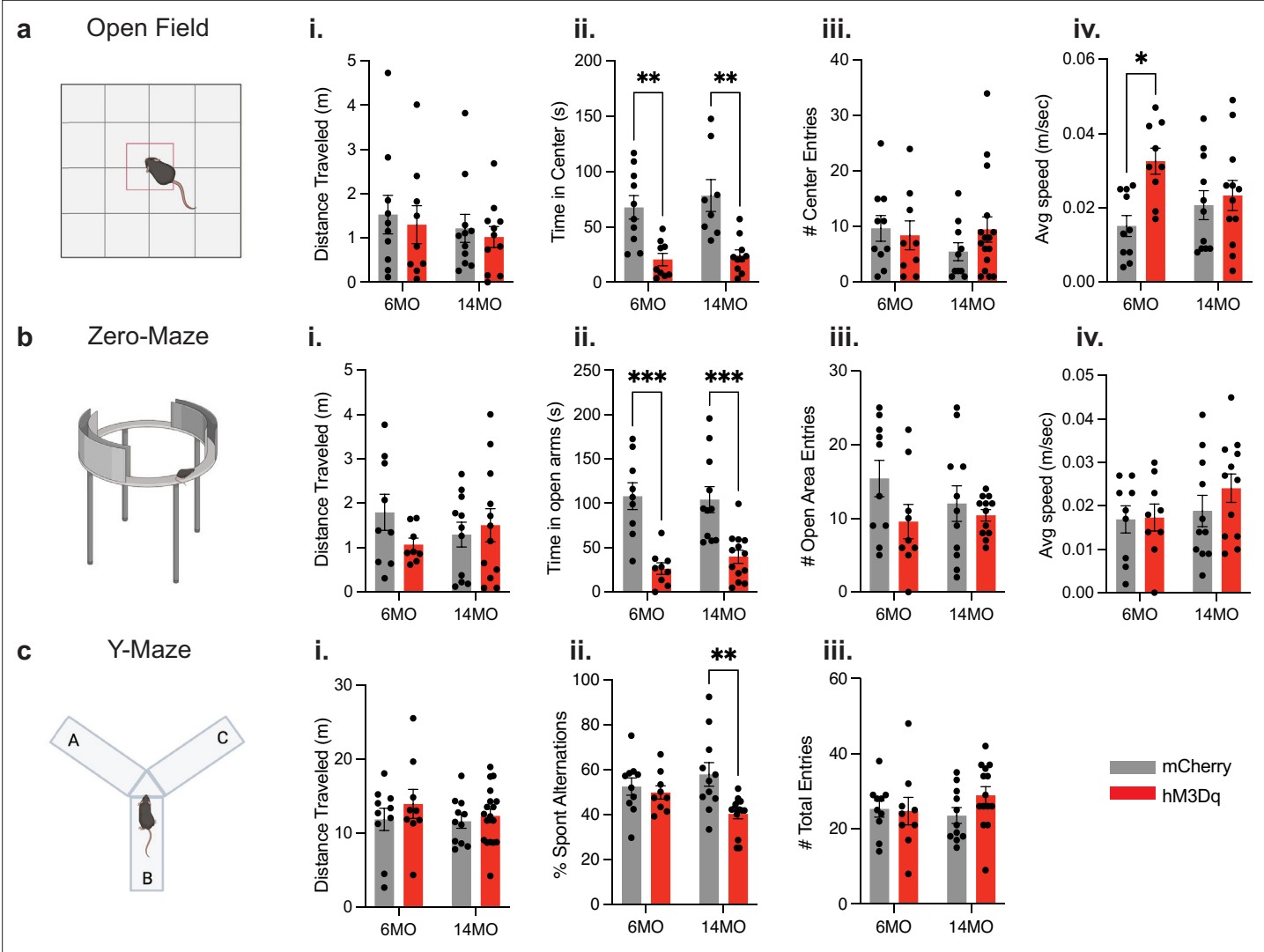

**Figure 3.** Chronic activation of an aversive engram induces behavioral changes in anxiety and working memory. (**a**) Schematic of open-field test; (**i**) total distance traveled, (**ii**) total time spent in center, (**iii**) number of entries into the center, and (**iv**) average speed in 6-month mCherry and hM3Dq and 14-month mCherry and hM3Dq after 3 months of stimulation. (**b**) Schematic of zero-maze test; (**i**) total distance traveled, (**ii**) total time spent in open arms, (**iii**) number of entries into the open area, and (**iv**) average speed. (**c**) Schematic of y-maze test; (**i**) total distance traveled, (**ii**) percentage of spontaneous alterations, and (**iii**) total number of entries into arms. Statistical measure used a two-way ANOVA with time point and group as factors. Tukey's post hoc tests were performed. Error bars indicate SEM. $p \leq 0.05$, **$p \leq 0.01$, ***$p \leq 0.001$, ****$p \leq 0.0001$, ns = not significant. n = 8–11 per group after outlier removal.

[vCA1] Interaction: $F_{(1,8)} = 2.824$, $p=0.1314$; age: $F_{(1,8)} = 0.5443$, $p=0.4817$; group: $F_{(1,8)} = 2.947$, $p=0.1244$) (*Figure 1—figure supplement 1e and f*).

To test the hypothesis that chronic activation of a negative memory is sufficient to increase anxiety-like behaviors, we subjected our mice to the open-field and zero-maze tests (*Figure 3*). In the open field (*Figure 3a*), 6- and 14-month-old hM3Dq mice spent less time in the center compared to age-matched mCherry mice (two-way ANOVA); interaction: $F_{(1,33)} = 0.1438$, $p=0.7070$; age: $F_{(1,33)} = 0.5858$, $p=0.4495$; group: $F_{(1,33)} = 29.56$, $p<0.0001$; Tukey's multiple comparisons [group]: 6MO mCherry vs. hM3Dq, $p=0.0049$; 14MO mCherry vs. hM3Dq, $p=0.0016$) (*Figure 3aii*). We observed no significant differences in total distance traveled across all groups (two-way ANOVA; interaction: $F_{(1,37)} = 0.0016$, $p=0.9675$; age: $F_{(1,37)} = 0.6862$, $p=0.4128$; group: $F_{(1,37)} = 0.3522$, $p=0.5565$) (*Figure 3ai*). Further, 6-month-old hM3Dq mice had an increase in average speed in the open field compared to mCherry mice (two-way ANOVA; interaction: $F_{(1,38)} = 4.003$, $p=0.526$; age: $F_{(1,38)} = 0.2346$, $p=0.6306$; group: $F_{(1,38)} = 7.306$, $p=0.0102$; (Tukey's [group]: 6MO mCherry vs. hM3Dq,

p=0.0150; 14MO mCherry vs. hM3Dq, p=0.9532) (*Figure 3aiv*). However, there were no differences in the number of center entries across all groups, indicating that mice in mCherry and hM3Dq groups are entering the center the same number of times, but spending less time per visit (two-way ANOVA; interaction: $F_{(1,41)}$ = 1.236, p=0.2728; age: $F_{(1,41)}$ = 0.4424, p=0.5097; group: $F_{(1,41)}$ = 0.3370, p=0.5648) (*Figure 3aiii*).

In the zero maze (*Figure 3b*), the 6- and 14-month-old hM3Dq mice spent less time in the center compared to age-matched mCherry mice (two-way ANOVA; interaction: $F_{(1,38)}$ = 0.5633, p=0.4575; age: $F_{(1,38)}$ = 0.1709, p=0.6816; group: $F_{(1,38)}$ = 40.22, p<0.0001; Tukey's [group]: 6MO mCherry vs. hM3Dq, p=0.0002; 14MO mCherry vs. hM3Dq, p=0.0007) (*Figure 3bii*). No significant differences were observed in the total distance traveled (*Figure 3bi*) or number of entries into the open area across all groups (two-way ANOVA; [total distance traveled] interaction: $F_{(1,36)}$ = 1.915, p=0.1749; age: $F_{(1,36)}$ = 0.0093, p=0.9237; group: $F_{(1,36)}$ = 0.6022, p=0.4428. [Open area entries] Interaction: $F_{(1,38)}$ = 1.082, p=0.3049; age: $F_{(1,38)}$ = 0.3839, p=0.5392; group: $F_{(1,38)}$ = 3.286, p=0.0778) (*Figure 3biii*). These findings indicate that mCherry and hM3Dq mice entered the open area the same number of times, but hM3Dq mice had shorter visits. We also observed no significant difference in average speed across all groups (two-way ANOVA; interaction: $F_{(1,37)}$ = 0.5118, p=0.4788; age: $F_{(1,37)}$ = 1.659, p=0.2057; group: $F_{(1,37)}$ = 0.7180, p=0.4023) (*Figure 3biv*). Ultimately, our observations from open field and zero maze support the hypothesis that chronic activation of a negative memory is sufficient to increase anxiety-like behaviors.

The mice also underwent y-maze testing in order to test the hypothesis that chronic activation of a negative memory is sufficient to impair spatial working memory (*Figure 3c*). In this task, mice were required to remember the most recent arm of the maze that was explored and subsequently navigate to a 'novel' arm. Mice freely explored the three arms of the maze (A, B, C) and spatial working memory was assessed by measuring the percentage of spontaneous alternations (i.e., ABC, CBA, CAB), or three consecutive entrances to a novel arm without repetition. A lower percentage of spontaneous alternations indicates a deficit in spatial working memory as they cannot recognize what arm they were in just moments prior. We observed a higher percentage of spontaneous alterations in 14-month-old mCherry than 14-month-old hM3Dq mice (two-way ANOVA); interaction: $F_{(1,40)}$ = 4.088, p=0.0499; age: $F_{(1,40)}$ = 0.3080, p=0.5820; group: $F_{(1,40)}$ = 7.517, p=0.0091; Tukey's [group]: 14MO mCherry vs. hM3Dq, p=0.0045 (*Figure 3cii*). However, no significant difference in the percentage of spontaneous alterations was observed between 6-month-old mCherry and hM3Dq mice (Tukey's [group]: 6MO mCherry vs. hM3Dq, p=0.9634) (*Figure 3cii*). Additionally, we observed no significant differences in the total distance traveled or total number of entries across all groups (two-way ANOVA; [total distance traveled] interaction: $F_{(1,44)}$ = 0.2668, p=0.6081; age: $F_{(1,44)}$ = 0.5106, p=0.4787; group: $F_{(1,44)}$ = 1.181, p=0.2831; *Figure 3ci*). [Total number of entries] Interaction: $F_{(1,40)}$ = 1.332, p=0.2553; age: $F_{(1,40)}$ = 0.2236, p=0.6389; group: $F_{(1,40)}$ = 0.8247, p=0.3692 (*Figure 3ciii*). Therefore, our observations suggest an inability of the mice to remember which arms of the y-maze they previously explored and may represent an impairment in working memory in the 14-month cohort.

Finally, mice had a 'negative' engram tagged in a fearful CxtA before the initiation of chronic activation (*Figure 4a*). During this CFC session, mCherry and hM3Dq mice in the 3- and 11-month groups displayed no differences in freezing behavior, as expected (two-way ANOVA; interaction: $F_{(1,42)}$ = 0.1342, p=0.7150; age: $F_{(1,42)}$ = 12.20, p=0.0011; group: $F_{(1,42)}$ = 2.913, p=0.0953) (*Figure 4ai–ii*). However, to test the hypothesis that chronic negative engram activation is sufficient to induce impaired fear memory recall and extinction, mice were subjected to remote memory recall after 3 months of treatment. The only significant difference in remote recall behavior was an increase in freezing between 6- and 14-month mCherry mice (two-way ANOVA; interaction: $F_{(1,41)}$ = 2.531, p=0.1193; age: $F_{(1,41)}$ = 10.08, p=0.0028; group: $F_{(1,41)}$ = 6.406, p=0.0153; Tukey's [age]: 6 vs. 14MO mCherry, p=0.0101; 6 vs. 14MO hM3Dq, p=0.6672. [Group] 6MO mCherry vs. hM3Dq, p=0.0449; 14MO mCherry vs. hM3Dq, p=0.8890) (*Figure 4bi–ii*). Mice typically display increased freezing behavior as they age, so these effects during remote recall are expected. For the extinction session in the same context, there were no significant differences across all groups at 6 and 14 months of age (*Figure 4ci–ii*). However, we observed a strong trend showing an increase in freezing in the 6-month-old hM3Dq mice that was like remote recall (two-way ANOVA; interaction: $F_{(1,41)}$ = 2.618, p=0.1133; age: $F_{(1,41)}$ = 2.335, p=0.1342; group: $F_{(1,41)}$ = 5.763, p=0.0210; Tukey's [group]: 6MO mCherry vs. hM3Dq, p=0.0590; 14MO mCherry vs. hM3Dq, p=0.9269) (*Figure 4di–ii*). This may suggest that these mice are displaying

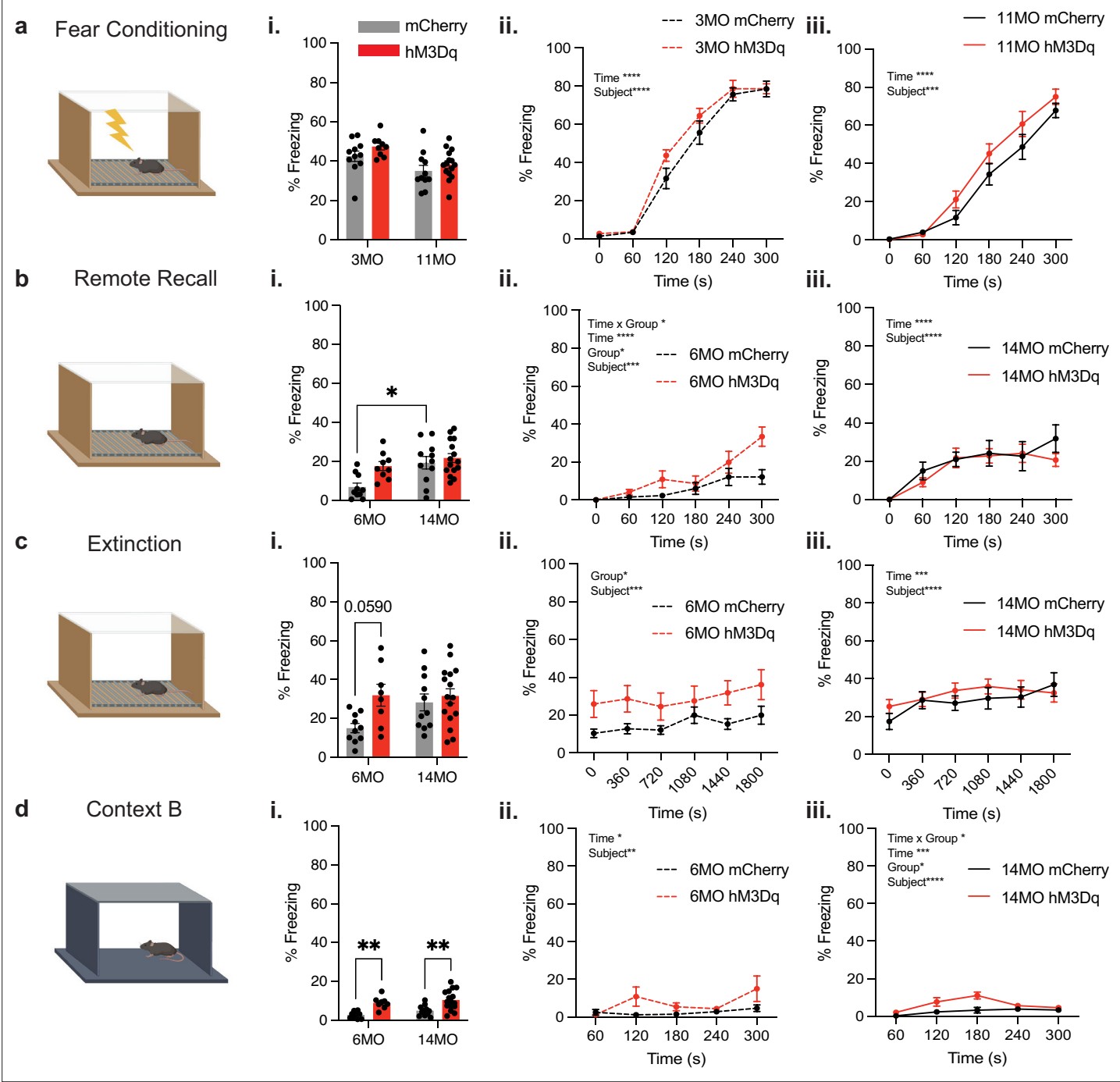

**Figure 4.** Chronic activation of the negative engram induces behavioral changes in fear memory. (**a**) Schematic of contextual fear conditioning (CFC) in context A wherein 3- and 11-month-old mice received four, 1.5 mA, 2 s foot shocks. Following the CFC session, the negative engram was tagged in all mice by intraperitoneal (IP) injection of 4-hydroxytamoxifen (4-OHT). (**i**) Average percent freezing levels and total percentage freezing across the 300 s CFC session for (**ii**) 3-month groups and (**iii**) 11-month groups (two-way ANOVA RM; [3 months] interaction: $F_{(5,130)} = 1.131$, $p=0.3473$; time: $F_{(5,130)} = 115.6$, $p<0.0001$; group: $F_{(1,26)} = 2.737$, $p=0.1101$; subject: $F_{(26,130)} = 3.056$, $p<0.0001$. [11 months] Interaction: $F_{(5,90)} = 1.218$, $p=0.3074$; time: $F_{(5,90)} = 234.4$, $p<0.0001$; group: $F_{(1,18)} = 1.862$, $p=0.1892$; subject: $F_{(18,90)} = 2.887$, $p=0.0005$. (**b**) Schematic of remote recall performed after 3 months of negative engram stimulation. Mice were returned to context A for 5 min in the absence of foot shocks. (**i**) Average percent freezing levels and total percentage freezing across the 300 s remote recall session for (**ii**) 6-month groups and (**iii**) 14-month groups two-way ANOVA RM; [6 months] interaction: $F_{(5,85)} = 15.72$, $p=0.0115$; time: $F_{(5,85)} = 15.72$, $p<0.001$; group: $F_{(1,17)} = 6.588$, $p=0.0200$; subject: $F_{(17,85)} = 2.458$, $p=0.0035$. [14 months] Interaction: $F_{(5,130)} = 0.9027$, $p=0.4815$; time: $F_{(5,130)} = 14.18$, $p<0.0001$; group: $F_{(1,26)} = 0.3967$, $p=0.5343$; subject: $F_{(26,130)} = 4.107$, $p<0.0001$. (**c**) Schematic of extinction wherein mice were placed in context A for 30 min in the absence of foot shocks. (**i**) Average percent freezing levels and total

*Figure 4 continued on next page*

*Figure 4 continued*

percentage freezing across the 1800s extinction session for (**ii**) 6-month groups and (**iii**) 14-month groups (two-way ANOVA RM; [6 months] interaction: $F_{(5,85)} = 0.4133$, p=0.8383; time: $F_{(5,85)} = 1.979$, p=0.0899; group: $F_{(1,17)} = 5.494$, p=0.0315; subject: $F_{(17,85)} = 7.352$, p<0.0001. [14 months] interaction: $F_{(5,130)} = 1,362$, p=0.2570; time: $F_{(5,130)} = 5.387$, p=0.0002; group: $F_{(1,26)} = 0.3836$, p=0.5411; subject: $F_{(26,130)} = 11.16$, p<0.0001). (**d**) Schematic of the novel context B used to assess generalization. (**i**) Average percent freezing levels and total percentage freezing across the 300 s generalization session for (**ii**) 6-month groups and (**iii**) 14-month groups (two-way ANOVA RM; [6 months] interaction: $F_{(4,72)} = 2.257$, p=0.0713; time: $F_{(4,72)} = 3.373$, p=0.0138; group: $F_{(1,18)} = 4.151$, p=0.0566; subject: $F_{(18,72)} = 2.537$, p=0.0028. [14 months] interaction: $F_{(4,100)} = 2.575$, p=0.0421; time: $F_{(4,100)} = 5.717$, p=0.0003; group: $F_{(1,25)} = 6.343$, p=0.0186; subject: $F_{(25,100)} = 3.175$, p<0.0001). Statistical analysis utilized a two-way ANOVA with time point and group as factors and two-way repeated measures (RM) ANOVA with time (seconds) and group as factors across 6- and 14-month-old mice. Tukey's or Sidak's post hoc tests were performed where applicable. Error bars indicate SEM. p≤0.05, **p≤0.01, ***p≤0.001, ****p≤0.0001, ns = not significant. n = 8–11 per group after outlier removal.

an increase in fear during remote recall of a fearful experience from 3 months prior and have difficulty appropriately decreasing levels of fear during extinction. Finally, mice were placed in a neutral, novel context B at this remote time point after chronic activation to test for generalization of fear. In this context B, mice in the hM3Dq groups at both 6 and 14 months of age displayed increased freezing in a neutral context compared to mCherry controls (two-way ANOVA; Iinteraction: $F_{(1,41)} = 0.09801$, p=0.7558; age: $F_{(1,41)} = 2.624$, p=0.1129; group: $F_{(1,41)} = 26.53$, p<0.0001; Tukey's [group]: 6MO mCherry vs. hM3Dq, p=0.0053; 14MO mCherry vs. hM3Dq, p=0.0026) (*Figure 4g and h*). Chronic negative engram activation induced selective deficits in extinction and modestly in fear generalization, which may point to the negative impact of fearful memories in human patients with disorders like PTSD (*Maren and Holmes, 2016*).

## Negative engram stimulation increases glial cell number and changes morphology

We next tested the hypothesis that chronic negative engram activation will induce glial changes related to inflammation by performing cell counts and morphological characterization of both microglia (Iba1+ ) and astrocytes (GFAP+ ) in the hippocampus. For microglia (*Figure 5a–j*, *Figure 5— figure supplement 1*), 14-month-old mice in the hM3Dq group displayed an increase in Iba1+ cells, while 6-month-old mice did not (two-way ANOVA; interaction: $F_{(1,8)} = 7.615$, p=0.0247; age: $F_{(1,8)} = 3.259$, p=0.1087; group: $F_{(1,8)} = 13.42$, p=0.0064; Tukey's [group]: 6MO mCherry vs. hM3Dq, p=0.9906; 14MO mCherry vs. hM3Dq, p=0.0113. [Age] 6 vs. 14MO mCherry, p=0.9876; 6 vs. 14MO hM3Dq, p=0.0704) (*Figure 5c*). Additionally, these 14-month-old hM3Dq mice had microglia with increased ramification index, cell territory volume, and minimum and maximum branch length, suggesting a distinct change in phenotype with negative memory activation (two-way ANOVA; [ramification index] interaction: $F_{(1,8)} = 1.908$, p=0.2046; age: $F_{(1,8)} = 1.950$, p=0.2001; group: $F_{(1,8)} = 15.11$, p=0.0046. Tukey's [group]: 6MO mCherry vs. hM3Dq, p=0.3513; 14MO mCherry vs. hM3Dq, p=0.03242 (*Figure 5d*). [Cell territory volume] interaction: $F_{(1,8)} = 3.494$, p=0.0965; age: $F_{(1,8)} = 0.05044$, p=0.8279; group: $F_{(1,8)} = 16.45$, p=0.0037. Tukey's [group]: 6MO mCherry vs. hM3Dq, p=0.4564; 14MO mCherry vs. hM3Dq, p=0.0129 (*Figure 5e*). [Min branch length] interaction: $F_{(A1,8)} = 8.287$, p=0.0206; age: $F_{(1,8)} = 9.619$, p=0.0146; group: $F_{(1,8)} = 7.470$, p=0.0257. Tukey's [group]: 6MO mCherry vs. hM3Dq, p=0.9996; 14MO mCherry vs. hM3Dq, p=0.0174. [Age]: 6 vs. 14MO mCherry, p=0.9985; 6 vs. 14MO hM3Dq, p=0.0141 (*Figure 5i*). [Max branch length] Interaction: $F_{(1,8)} = 4.094$, p=0.0777; age: $F_{(1,8)} = 1.214$, p=0.3026; group: $F_{(1,8)} = 8.964$, p=0.0172. Tukey's [group]: 6MO mCherry vs. hM3Dq, p=0.8994; 14MO mCherry vs. hM3Dq, p=0.0309 (*Figure 5j*)). However, there were no significant differences in the average centroid distance, number of endpoints, and average branch length for Iba1+ cells (two-way ANOVA; [Avg centroid distance] interaction: $F_{(1,8)} = 0.7707$, p=0.4056; age: $F_{(1,8)} = 1.155$, p=0.3138; group: $F_{(1,8)} = 5.459$, p=0.0477. Tukey's [group]: 6MO mCherry vs. hM3Dq, p=0.7370; 14MO mCherry vs. hM3Dq, p=0.1838; *Figure 5f*. [Number of endpoints] Interaction: $F_{(1,8)} = 2.110$, p=0.1884; age: $F_{(1,8)} = 2.158$, p=0.1801; group: $F_{(1,8)} = 0.0237$, p=0.8815; *Figure 5g*. [Avg branch length] Interaction: $F_{(1,8)} = 0.7574$, p=0.4095; age: $F_{(1,8)} = 0.02109$, p=0.8881; group: $F_{(1,8)} = 0.1211$, p=0.7369 (*Figure 5g*)).

For astrocytes (*Figure 5k–t*, *Figure 5—figure supplement 1*), there was a similar increase in the number of GFAP+ cells in the hippocampus, but in both the 6- and 14-month groups receiving hM3Dq activation (two-way ANOVA; interaction: $F_{(1,8)} = 0.05526$, p=0.8201; age: $F_{(1,8)} = 6.824$,

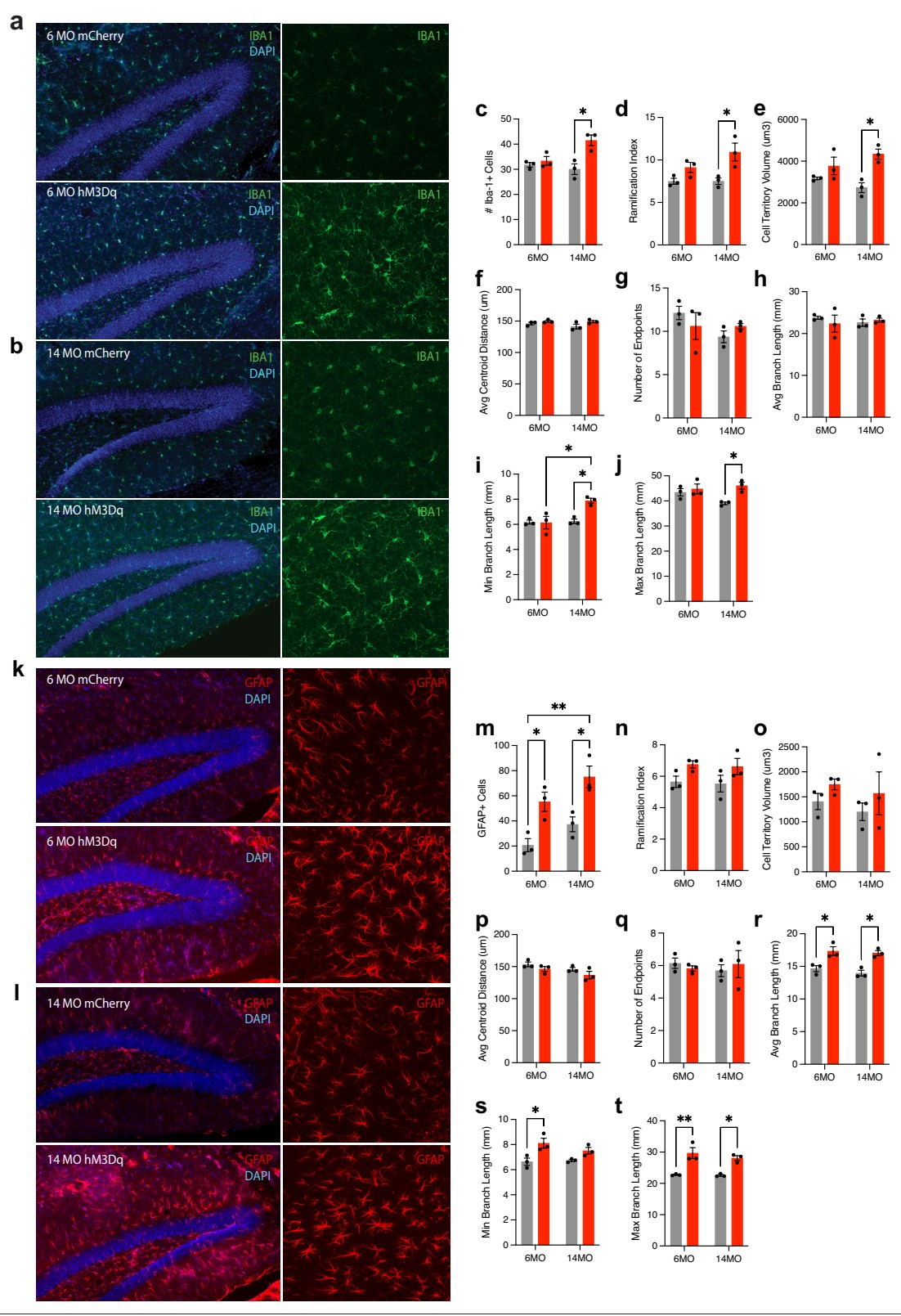

**Figure 5.** Chronic modulation of the negative engram induces changes in the number and morphology of microglia and astrocytes. (**a, b**) Representative images of Iba1+ microglia (green) in the hippocampus of (**a**) 6-month mCherry and hM3Dq mice and (**b**) 14-month mCherry and hM3Dq mice following 3 months of stimulation. (**c**) Number of Iba1+ cells. (**d–j**) Quantified morphological analysis of Iba1+ cells; (**d**) ramification index, (**e**) cell territory volume, (**f**) average centroid distance, (**g**) number of endpoints, (**h**) average branch length, (**i**) minimum branch length, and (**j**) maximum

*Figure 5 continued on next page*

*Figure 5 continued*

branch length. (**k, l**) Representative images of GFAP+ astrocytes (red) in the hippocampus of (**k**) 6-month mCherry and hM3Dq mice and (**l**) 14-month mCherry and hM3Dq mice following 3 months of stimulation. (**m**) Number of GFAP+ cells. (**n–t**) Quantified morphological analysis of GFAP+ cells; (**n**) ramification index, (**o**) cell territory volume, (**p**) average centroid distance, (**q**) number of endpoints, (**r**) average branch length, (**s**) minimum branch length, and (**t**) maximum branch length. Statistical analysis used a two-way ANOVA with time point and group as factors. Tukey's post hoc tests were performed. Error bars indicate SEM. $p \leq 0.05$, **$p \leq 0.01$, ***$p \leq 0.001$, ****$p \leq 0.0001$, ns = not significant. For cell counts, n = 3 mice/group × 18 single-tile ROIs each (GFAP, Iba1). For morphology, n = 3 mice/group × 3 single-tile ROIs each (GFAP, Iba1), each containing 20–30 glial cells.

The online version of this article includes the following figure supplement(s) for figure 5:

**Figure supplement 1.** Analysis of astrocyte and microglial morphology using the MATLAB-based tool 3DMorph.

p=0.0310; group: $F_{(1,8)}$ = 26.95, p=0.0008; Tukey's [group]: 6MO mCherry vs. hM3Dq, p=0.0328; 14MO mCherry vs. hM3Dq, p=0.0208) (*Figure 5m*). Additionally, these astrocytes at both ages had increased average branch length and maximum branch length (two-way ANOVA; [Avg branch length] interaction: $F_{(1,8)}$ = 0.1537, p=0.7052; age: $F_{(1,8)}$ = 0.9883, p=0.3493; group: $F_{(1,8)}$ = 32.75, p=0.0004; Tukey's [group]: 6MO mCherry vs. hM3Dq, p=0.0228; 14MO mCherry vs. hM3Dq, p=0.0109; *Figure 5r*. [Max branch length] Interaction: $F_{(1,8)}$ = 0.5438, p=0.4819; age: $F_{(1,8)}$ = 0.8447, p=0.3849; group: $F_{(1,8)}$ = 41.82, p=0.0002. Tukey's [group]: 6MO mCherry vs. hM3Dq, p=0.0041; 14MO mCherry vs. hM3Dq, p=0.0155) (*Figure 5t*). Astrocytes in the 6-month hM3Dq group additionally had increased minimum branch length (two-way ANOVA; interaction: $F_{(1,8)}$ = 1.568, p=0.2459; age: $F_{(1,8)}$ = 0.7927, p=0.3993; group: $F_{(1,8)}$ = 16.26, p=0.0038; Tukey's [group]: 6MO mCherry vs. hM3Dq, p=0.0238; 14MO mCherry vs. hM3Dq, p=0.2759) (*Figure 5s*). However, astrocytes did not show any significant differences for any group in their ramification index, cell territory volume, average centroid distance, and number of endpoints (two-way ANOVA, [ramification index] interaction: $F_{(1,8)}$ = 0.0003, p=0.9856; age: $F_{(1,8)}$ = 0.08582, p=0.7770; group: $F_{(1,8)}$ = 6.677, p=0.0324. Tukey's [group]: 6MO mCherry vs. hM3Dq, p=0.3233; 14MO mCherry vs. hM3Dq, p=0.3339; *Figure 5n*. [Cell territory volume] Interaction: $F_{(1,8)}$ = 0.0028, p=0.9584; age: $F_{(1,8)}$ = 0.5819, p=0.4674; group: $F_{(1,8)}$ = 1.996, p=0.1954; *Figure 5o*. [Avg centroid distance] Interaction: $F_{(1,8)}$ = 0.04699, p=0.8338; age: $F_{(1,8)}$ = 4.275, p=0.0725; group: $F_{(1,8)}$ = 3.856, p=0.0852; *Figure 5p*. [Number of endpoints] Interaction: $F_{(1,8)}$ = 0.5463, p=0.4810; age: $F_{(1,8)}$ = 0.0365, p=0.8532; group: $F_{(1,8)}$ = 0.0071, p=0.9346) (*Figure 5q*). In summary, chronic activation of a negative engram induces an increase in glial cell number and changes their morphology in both young and older cohorts.

## Discussion

We chronically activated negative engrams in the vHPC using chemogenetics over the course of 3 months. This stimulation protocol induced detrimental behavioral effects as evidenced by increased anxiety-like behaviors, decreased working memory, and increased fear responses across extinction and generalization. Notably, our chronic negative engram activation significantly impacted microglial and astrocytic number and morphology within the hippocampus. Together, these findings present evidence for the role of negative memories in producing maladaptive cellular and behavioral responses within a rodent model and further our understanding of the broad impacts of chronic negative thinking-like behaviors that may detrimentally affect human health.

### Molecular features of engram cells

First, to characterize the molecular features of engram cells, we performed RNA-sequencing and utilized GSEA to compare gene expression profiles in cells that were labeled during a positive or negative experience (*Rao-Ruiz et al., 2019*; *Fuentes-Ramos et al., 2021*; *Sardoo et al., 2022*). Using a priori gene sets of interest, we found that negative engram cells upregulated genes that were associated with pathways involved in neurodegeneration, proinflammatory processes, and apoptosis. In contrast, positive engram cells upregulated genes involved in neuroprotective and anti-inflammatory processes. Notable genes that were significantly upregulated in the negative engram cells were secreted phosphoprotein 1 (*Spp1*) and transthyretin (*Ttr*). SPP1, also known as osteopontin, is an extracellular matrix protein widely expressed in immune cells, such as macrophages and microglia, mediating inflammatory responses. Blocking SPP1 protein expression in cells within the hippocampus has been shown to attenuate inflammation-induced depressive-like behaviors in mice (*Li et al., 2023*).

Furthermore, the *Spp1* gene is consistently upregulated after chronic social stress (*Stankiewicz et al., 2015*), recall of contextual fear (*Barnes et al., 2012*), and in rats with increased trait anxiety (*Díaz-Morán et al., 2013*). Mice receiving chronic negative memory stimulation in our study displayed similar increases in anxiety-like behaviors and impairments in fear responses. Further, TTR is a serum transporter for thyroid hormones and retinol binding proteins that has been shown to be detrimental when misfolded and aggregated in various forms of amyloidosis, while also having evidence of protective effects in Alzheimer's disease (AD) (*Buxbaum et al., 2008*; *Sharma et al., 2019*). Interestingly, TTR mRNA and protein were increased in both male and female rats that underwent acute or chronic stress, emphasizing the potential impact of glucocorticoid hormones on TTR protein levels (*Martinho et al., 2012*). Together, these findings suggest that genes upregulated in negative engram cells are linked to stress-induced changes in anxiety, fear, and depressive-like behaviors.

Strikingly, negative engram cells also significantly upregulated amyloid precursor protein (*App*) and apolipoprotein E (*Apoe*) genes. APP plays a central role in AD pathology through generation of amyloid plaques (*Lee et al., 2018*; *Lee et al., 2020*). *Apoe* genotype accounts for the majority of AD risk and pathology (*Johnson et al., 2015*; *Schuff et al., 2009*), with the E4 isotype accounting for as much as 50% of AD in the United States (*Raber et al., 2004*). The upregulation of these genes involved in neurodegeneration and neuroinflammation is in line with human research showing a link between repetitive negative thinking and increased risk of cognitive decline and vulnerability to dementia (*Schlosser et al., 2020*). Furthermore, negative engram cells downregulated neuroprotective and anti-inflammatory genes, notably brain-derived neurotrophic factor (*Bdnf*). BDNF is a key molecule involved in memory in the healthy and pathological brain. Interventions such as exercise and antidepressant treatments enhance the expression of BDNF protein (*Miranda et al., 2019*). Future studies may further investigate the neuroprotective role of positive engrams to test, for instance, if activating a positive engram in a mouse model of AD confers protective effects on the brain and slows the progression of disease. Indeed, evidence of positive memory impact on mice with a depression-like phenotype has already shown promising results in this area of research (*Ramirez et al., 2015*). Together, these genetic data begin to uncover the molecular landscape that differentiates positive and negative memory-bearing cells, linking them to either protective or pathological brain states, respectively.

## Chronic stimulation of engrams alter behavior and cellular responses

Chronic modulation of engram cells in rodents has differential effects of behavior, depending on the brain region and valence of the tagged neurons. Reactivation of positive engram cells in the hippocampus rescued stress-induced depression-like phenotype and increased neurogenesis in mice (*Ramirez et al., 2015*). Further, chronic stimulation of dorsal or ventral hippocampal fear engrams produced a context-specific reduction or enhancement of fear responses, respectively (*Chen et al., 2019*). Interestingly, chronic optogenetic activation of dorsal hippocampal fear engrams results in context-specific reduction in fear expression that resembles extinction, even in mice with impaired extinction due to ethanol withdrawal (*Cincotta et al., 2021*). Activation of negative engram cells labeled during chronic social defeat stress induced social avoidance, a behavior common in human patients with MDD (*Zhang et al., 2019*). Together, these studies suggest that activity-dependent modulation of hippocampal cells over prolonged periods of time is sufficient to modify behavioral responses and brain states in mice.

Engram studies have predominantly focused on the neuronal phenotypes resulting from activity-dependent manipulations; however, glial cell responses to chronic memory activation and subsequent behavioral changes are unknown. Microglia are the brain's resident immune cells, actively surveying the environment to maintain homeostasis and protect against insult and disease. Under normal conditions, microglial cells display small cell bodies and fine, highly ramified processes that allow them to detect and respond to threats (*Nimmerjahn et al., 2005*). Once a threat is detected (i.e., pathogens, debris, injury), microglia retract and exhibit large soma size and thickening, entering an 'ameboid' state. Evidence has accumulated that microglial morphology may be more complex than this dichotomy between 'surveying' and 'ameboid' microglia (*Vidal-Itriago et al., 2022*; *Leyh et al., 2021*; *Reddaway et al., 2023*). Our chronic negative engram activation may induce an 'insult' on the brain, as evidenced by an increase in Iba1 expression and hyper-ramified morphology, indicating a complex change in microglial phenotype. In support of these findings, chronic stress similarly induced

an increase in Iba1 expression and hyper-ramified microglia in the prefrontal cortex (PFC) that was correlated with impaired spatial working memory (*Hinwood et al., 2013*). Further supporting these findings, a mouse model of PTSD displayed a similar increase in Iba1+ and hyper-ramified microglial cells in the PFC and CA1 layer of the hippocampus (*Smith et al., 2019*). Together, the changes in microglial cell number and morphology observed in the context of chronic negative engram stimulation are reminiscent of the alterations seen in chronic stress, PTSD, and related depression-like behaviors.

Further, astrocytes express glial fibrillary acidic protein (GFAP), a critical intermediate filament responsible for maintaining their cytoskeletal structure (*Yang and Wang, 2015*). Chronic restraint stress has been associated with static GFAP+ astrocytic number and atrophy of astrocytic processes in the hippocampus, amygdala, and retrosplenial cortex (*Kassem et al., 2013*). Studies have primarily shown a decrease in GFAP+ astrocyte numbers and atrophy in rodent brains subjected to chronic and acute stressors (*Tynan et al., 2013*; *Banasr and Duman, 2008*; *Gosselin et al., 2009*; *Miguel-Hidalgo et al., 2000*; *Bender et al., 2016*; *Codeluppi et al., 2021*). However, recent studies have found a variety of changes in astrocytic morphology and number depending on the type of stressor and brain region. Conversely, others report an increase in GFAP+ cells in the hippocampus (*Jang et al., 2008*). Additionally, unpredictable chronic mild stress substantially increased GFAP immunoreactivity and promoted a more branched, reactive morphology in the hippocampus, which was correlated with increases in anxiety and anhedonic-like behaviors (*Du Preez et al., 2021*). This observed phenotype was coupled with similar changes in Iba1+ microglia, featuring increased branching complexity and ramification, longer processes, and increased volume of occupation, as discussed previously (*Du Preez et al., 2021*; *Hinwood et al., 2013*). In the case of humans, the discourse on glial changes in chronic stress and depression-like behaviors has been similarly complex. GFAP concentrations in the cerebellum, prefrontal cortex, hippocampus, and anterior cingulate cortex have been reported to decrease in patients with major depression compared to controls (*Fatemi et al., 2004*; *Si et al., 2004*; *Webster et al., 2005*; *Cobb et al., 2016*). However, a recent study found increased GFAP concentrations in the cerebral spinal fluid of patients with major depression compared to controls (*Michel et al., 2021*). Our findings that astrocytes increase in number and branch length in the hippocampus after chronic negative memory stimulation mirror the impacts of chronic stress on glial morphology. Further research is needed to understand these nuanced changes across brain regions, animal models, and type of stressors and influence on behavior.

Our previous work demonstrated that global chronic activation of vHPC excitatory neurons or astrocytes for three months was sufficient to induce a variety of behavioral and cellular changes (*Suthard et al., 2023*). In the current study, we modulated hippocampal cells in an activity-dependent manner, gaining access to the specific impact of chronic negative memory activation on behavioral and cellular responses. Collectively, our data show a similar impact of CaMKII+ neuron and negative fear memory activation on fear extinction behaviors – namely, an increase in freezing responses. However, GFAP+ astrocyte activation induced a mild decrease in anxiety-related behaviors in the open field, while our negative memory activation significantly increased anxiety. Histologically, our previous work had mild impacts on glial number and morphology with non-memory-specific neuron or astrocyte chronic activation (*Suthard et al., 2023*). The current study demonstrates a much more potent increase in both microglia and astrocyte number, as well as robust changes in morphological characteristics like ramification index, cell territory volume, and branch lengths, which underscore the efficacy of chronic negative engram stimulation. A concern, however, is that activation of vHPC neurons could induce local tissue pathology as evidenced by inflammation and glial response. As this previous work activated all CaMKII+ neurons in vHPC and induced very modest changes in glial cells, we expect that in the current work, stimulating a smaller ensemble of cells would induce less of such 'off-target' inflammation. Together, these findings highlight the differences between modulating cells in a cell-type-specific or activity-dependent manner (e.g., vHPC neurons or negative engram cells) and the resulting phenotypes that emerge.

Previous work from our lab and others has shown that there are differential effects of chronic activation of negative, positive, and neutral ensembles within the hippocampus. As discussed earlier, chronic reactivation of a positive valence engram was sufficient to reverse the effects of stress on depression-like behaviors, whereas modulating a neutral ensemble did not (*Ramirez et al., 2015*). Here, we speculate that chronic activation of a positive engram in healthy, unstressed mice may

produce behavioral outcomes similar to those effects seen in chronic environmental enrichment, with improved cognitive flexibility, memory, and even enhanced levels of anti-inflammatory proteins like BDNF (see *Bayne, 2018* for review). We further speculate that unlike neutral or negative memories, chronic stimulation of a positive memory may provide prophylactic effects against stress, degeneration, and aging, though this work remains an opportunity for future experiments. Other studies utilizing repeated negative memory activation (*Chen et al., 2019*; *Zhang et al., 2019*; *Cincotta et al., 2021*) showed content- and context-specific effects, such as bi-directional control over the strength of a fear memory in healthy and addiction-related states, with no changes observed in a neutral control group. Although our experiments here do not compare the effects of positive or neutral memory stimulation directly but rather compare a negative memory group to a fluorophore-only control, which is therefore a current limitation, we believe that chronic stimulation of each would produce unique sets of cellular and behavioral phenotypes that raise the intriguing notion of manipulating memories to mimic or ameliorate various pathological states.

## Relevance of chronic negative thinking-like behaviors to human health

Chronic memory activation may mimic symptoms of rumination, negative thinking, and ongoing worry shared by individuals with GAD, MDD, and PTSD (*Brewin et al., 2010*; *Newman et al., 2013*; *Watkins, 2008*; *McEvoy et al., 2013*), but the mechanisms behind this connection are currently unknown. In the healthy brain, 'inhibitory control' of the prefrontal cortex on the hippocampus is thought to keep these symptoms at bay. Recent work has suggested a framework where stress may compromise GABAergic signaling in the hippocampus (*Alaiyed et al., 2020*; *Banasr et al., 2017*; *Albrecht et al., 2021*), allowing a state of hippocampal hyperactivity to persist. This elevated hippocampal activity and dysfunction of GABAergic interneuron is evident in human patients with major depression and PTSD, and the activity levels are correlated with behavioral rumination and flashback intensity (*Osuch et al., 2001*; *Rayner et al., 2016*). Human functional imaging work has supported this framework by showing that fronto-hippocampal inhibitory control via GABAergic signaling underlies this inability to suppress unwanted thoughts. Further, GABAergic signaling in the hippocampus has been shown to produce impaired extinction of conditioned fear (*Müller et al., 2015*) and increased anxiety (*Crestani et al., 1999*). This mirrors our findings of increased anxiety and decreased extinction ability in mice that experienced negative memory activation. Further exploration into how chronic engram activation impacts excitatory-inhibitory (E/I) imbalance may provide a link between memory modulation and induction of effects on the brain similar to those related to rumination, negative thinking, and worry.

In humans, persistent worrying and rumination behaviors that are common in neuropsychiatric disorders can affect the health of the entire body. In the context of our findings, the concept of allostatic load refers to the 'wear and tear' the body experiences as a result of chronic exposure to stressful events. While the body attempts to produce a physiologically adaptive response through the process of allostasis, it is proposed that repeated cycles of allostasis in response to chronic stressors results in allostatic overload. In this state, the cumulative impact of chronic stress has overcome the body's ability to cope, ultimately culminating in disease (*McEwen and Stellar, 1993*; *McEwen, 2004*). This applies to the nervous system, wherein rumination has been shown to increase the risk of alcohol and substance abuse, especially in young individuals, as a form of compensatory behavior to numb the persistent negative thoughts (*Nolen-Hoeksema and Watkins, 2011*). Rumination after a stressful event is also linked to insomnia, poor sleep quality, and pre-sleep intrusive thoughts, as well as an increase in the risk of heart disease and cardiovascular issues via autonomic dysregulation (*Guastella and Moulds, 2007*; *Busch et al., 2017*). Together, we hope that the current study provides a novel framework in which memory itself may affect, and lead to, the maladaptive brain and body states described here.

## Chronic engram reactivation and its differing effects on freezing behavior

Previous work showed that experimental activation of recall-induced populations in the dentate gyrus leads to fear attenuation (*Khalaf et al., 2018*), while our experimental manipulation produces lasting negative impacts on the brain and behavior. Specifically, this study posits that this fear attenuation occurs in neuronal ensembles through updating or unlearning of the original memory trace via the engagement, rather than suppression, of an original traumatic experience. Notably, one major

difference is that our experiment reactivated an ensemble that was tagged during memory encoding, while they activated a remote recall-induced ensemble that was tagged 1 month after encoding. Although there is high overlap between the encoding and recall ensembles when mice are exposed to the conditioning context, these ensembles are not identical and may result in different behavioral phenotypes when chronically reactivated. Further, Khalaf et al. relied on reactivation of the recall-induced ensemble during extinction to facilitate rapid fear attenuation. This differs experimentally from our current work as their reactivation occurred during the extinction process in the previously conditioned context, while we reactivated chronically in the animal's home cage over the course of a longer time period. It may be necessary that the memory is reactivated, and thus, more liable to re-contextualization, in the original context compared to an unrelated home cage environment where there are no related cues present. Importantly, this previous work tested the attenuation of fear shortly after an extinction process, while we did not extinguish the memory with aid of the memory reactivation. Finally, we tested remote recall (i.e., 3 months post-conditioning), as opposed to a shorter time interval (28 days). Despite these discrepancies, it is indeed possible that our chronic reactivation of a fear acquisition ensemble over time mildly attenuates or changes the original memory in some way. Future work could build on the findings of Khalaf et al. and/or show differences in the activation of a conditioning vs. recall-induced ensemble on fear attenuation during extinction.

## Limitations of the study

### Limitations of chronic chemogenetic modulation

In this study, we rely on the successful chemogenetic modulation of fearful neuronal ensembles to assess its impact on behavior over a long period of time. This strategy requires that our ligand, DCZ, maintains an effect on neuronal excitability across our 3 months of chronic reactivation. Previous work has shown that CNO administration in drinking water over 1 month consistently inhibited hM4Di+ neurons without altering baseline neuronal excitability as measured by firing rate and potassium currents (*Xia et al., 2017*). Although this is only for 1 month, it is administered via the same oral route as our DCZ protocol and suggests that at least for that amount of time we are likely producing similar effects. However, this must be experimentally tested as CNO and DCZ are unique ligands for chemogenetic modulation and must be extended out from 1 month to 3 months to assess the longevity of its effects in the brain. If DCZ is only having an effect for 1 vs. 3 months, we are still observing enduring changes that resulted from this shorter-term disturbance, which nonetheless informs our understanding of negative memory's effects on the brain and behavior. On this same note, assessment of DREADD impact on E/I balance must be quantified utilizing both excitatory and inhibitory markers in future work to understand how chemogenetic modulation of engram cells may truly impact circuit-level E/I dysfunction.

Related to this, other work has shown evidence of DREADD toxicity at high titer levels of AAV2/7-CaMKII-hM4Di-mCherry in the HPC at 5 weeks (*Goossens et al., 2021*). We mitigated this risk by utilizing a viral strategy that targets a smaller number of engram cells using the AAV9-DIO-Flex-hM3Dq-mCherry virus at a lower titer, unlike this previous work that targeted all CaMKII+ cells in the HPC. Despite these efforts, we wanted to assess signs of decreasing neuronal health that may have resulted from chronic DREADD expression and utilized NeuN counts within multiple hippocampal subregions (*Yousef et al., 2017*). Here, we note that we did not observe significant decreases in NeuN across active Gq receptor or mCherry control groups in either of the age groups (*Figure 1—figure supplement 1*). However, immunohistochemistry using an individual marker may not be sufficient to capture the entire health profile of an individual neuron and future work should consider other markers of cell death or inflammation to ensure prolonged health of DREADD+ cells.

### Limitations of genetic tagging strategies

Further, the present study relies on use of the Tet-tag and TRAP2 genetic strategies to label engram cells for sequencing (Tet-tag) and chronic activation (TRAP2). While both systems rely on the *cFos* promoter to drive an effector of interest (fluorophore or DREADD), their efficacy and temporal resolution vary substantially depending on the genetic cell type, brain region, temporal parameters of Dox or 4-OHT delivery, subject-by-subject metabolic variability, and threshold to Dox induction given the promoter sequences inherent to each system. The TRAP2 line labels a subset of endogenously activeCA1 pyramidal cells (e.g., 5–18%) while the DOX system labels 20–40% of CA1 pyramidal cells

(*DeNardo et al., 2019*; *Monasterio et al., 2024*). Further, the temporal windows for each system range from hours in TRAP2, compared to 24–48 hr in the Tet-tag system, resulting in potentially less-specific labeling in the latter (*DeNardo et al., 2019*; *Denny et al., 2014*; *Liu et al., 2012*). Further, the efficacy of tagging a population of cells with either system will inherently constrain the number of cells that may overlap with cFos+ cells upon re-exposure to a given experience, given the difference in the percentage of initial cells labeled (*Kim and Cho, 2020*; *Ortega-de San Luis et al., 2023*; *Shpokayte et al., 2022*). However, it is important to note that tagging vHPC cells with both strategies is nonetheless sufficient to drive behavioral responses (*Shpokayte et al., 2022*; *Ortega-de San Luis et al., 2023*).

Finally, and promisingly, as more studies continue to link the in vivo physiological dynamics of these cell populations tagged using each system (e.g., compare *Pettit et al., 2022* with *Tanaka et al., 2018*) and correlating their activity to behavioral phenotypes, our field is in the prime position to uncover deeper principles governing hippocampus-mediated engrams in the brain. Together, we believe a more comprehensive understanding of these systems is fully warranted, especially in the service of further cataloging cellular similarities and differences within such tagged populations.

### Limitations of GSEA within engram cells

Although the gene sets were pre-selected to assess changes canonically involved in 'neurodegeneration' or 'neuroprotection' such as *App, Apoe, or Bdnf*, a major limitation of this technique is that we must avoid making strong claims about the actual function of these up- or downregulated genes without performing proper causal studies. We hope these findings provide an unbiased inventory of the changes induced within engram cells related to memory valence that can inspire future work testing the knock-in or knock-down of these genes to assess their direct impact on the brain and behavior.

### Limitations of immunohistochemistry analysis

Despite our study's assessment of changes in astrocytic and microglial morphology and number after our manipulation, using immunohistochemistry does not allow us to capture all changes that may be occurring at the gene or protein level, painting an incomplete picture of the impact on glial cells. Future work utilizing transcriptomic or proteomic profiling of both engram and non-neuronal cells will help the field understand how these cell types are interacting with one another and provide a more complete mechanistic understanding of these glial-related changes.

## Materials and methods
### Generation of single-cell suspension from mouse hippocampal tissue

As shown in *Figure 1* (RNA-sequencing), C57BL/6J mice were bilaterally injected with a 1:1 viral cocktail of AAV9-c-Fos-tTA and AAV9-TRE-eYFP into the vHPC. This activity-dependent labeling strategy captures sufficiently active neurons expressing the immediate-early gene, *cfos*. This system couples the *cfos* promoter to the expression of the tetracycline transactivator (tTA), which binds to the tetracycline response element (TRE) in its protein form. When DOX is present, such as in the animal's diet, this inhibits the binding of tTA to TRE, preventing eYFP labeling of active cells (*Liu et al., 2012*).

Prior to surgery, mice were placed on a Dox diet to inhibit the labeling of active neurons. After surgery and viral expression, on day 10, mice were taken off of Dox prior to engram tagging to open the labeling window. On day 11, mice were subjected to CFC (negative) or male-to-female interaction (positive) to label active neurons with eYFP. Mice were placed immediately back on Dox to close this tagging window. Then, 24 hr later, mice were sacrificed and brains were rapidly extracted, and the hippocampal regions were isolated by microdissection. Eight mice were pooled by each experimental condition. Single-cell suspension was prepared according to the guideline of the Adult Brain Dissociation Kit (Miltenyi Biotec, Cat# 13-107-677). Briefly, the hippocampal samples were incubated with digestion enzymes in the C Tube placed on the gentleMACS Octo Dissociator with Heaters with gentleMACS Program: 37C_ABDK_01. After termination of the program, the samples were applied through a MACS SmartStrainer (70 µm). Then a debris removal step and a red blood cell removal step were applied to obtain single-cell suspension. Viral injection surgery through RNA-sequencing was

performed in *Shpokayte et al., 2022*, and this dataset was utilized for further analysis in the current study.

## Isolation of EYFP-positive single cell by FACS

The single-cell suspension was subject to a BD FACSAria cell sorter according to the manufacturer's protocol to isolate eYFP-single cell population. This was performed in *Shpokayte et al., 2022*.

## Preparation of RNA-seq library

The RNA of FACS isolated eYFP-positive cells was extracted by using TRIzol (Invitrogen Life Technologies, MA) followed by Direct-zol kit (Zymo Research, CA) according to the manufacturer's instructions. Then the RNA-seq library was prepared using SMART-Seq v4 Ultra Low Input RNA Kit (TaKaRa Bio, USA). This was performed in *Shpokayte et al., 2022*.

## Analysis of RNA-seq data

The original RNA sequencing data was from *Shpokayte et al., 2022*. To analyze this data, the resulting reads from Illumina had good quality by checking with FastQC. The first 40 bp of the reads were mapped to mouse genome (mm10) using STAR (*Dobin et al., 2013*), which was indexed with Ensembl GRCm38.91 gene annotation. The read counts were obtained using featureCounts (*Liao et al., 2014*) function from the Subread package with an unstranded option. Reads were normalized by library size and differential expression analysis based on negative binomial distribution was done with DESeq2 (*Love et al., 2014*). Genes with FDR-adjusted p-value<0.00001 plus more than fourfold difference were considered to be differentially expressed. Raw data, along with gene expression levels, are being deposited to the NCBI Gene Expression Omnibus (*Liu and Yuan, 2022*). For Gene Ontology analysis, we ranked protein-coding genes by their fold changes and used them as the input for the GseaPreranked tool.

## GSEA

To further analyze the dataset generated in *Shpokayte et al., 2022*, we performed further analyses using GSEA (https://www.gsea-msigdb.org/gsea/index.jsp; *Subramanian et al., 2005*; *Mootha et al., 2003*). This is a computational method that determines whether a priori-defined set of genes shows statistically significant, concordant differences between two biological states. GSEA is a joint project of UC San Diego and Broad Institute. Gene sets were selected based on existing literature around genes important in neurodegeneration, inflammation, and apoptosis.

## Subjects (Figures 2–5)

Fos-based transgenic mouse, Fos-tm2.1(iCre-ERT2) raised on a C57BL/6 background (TRAP2) (Jackson Laboratories; strain #030323) mice were bred in-house. The mice were housed in groups of 2–5 mice per cage. The animal facilities (vivarium and behavioral testing rooms) were maintained on a 12:12 hr light cycle (07:00–19:00). Mice received food and water ad libitum before surgery. Following surgery, mice were group-housed with littermates and allowed to recover for a minimum of 4 weeks before experimentation for viral expression. All subjects were treated in accordance with protocol 201800579 approved by the Institutional Animal Care and Use Committee (IACUC) at Boston University.

## DCZ administration

All animals had their home cage water supply replaced with water-soluble DCZ in distilled water (diH$_2$O) every 5 days (HelloBio, #HB9126). Administration was calculated based on each mouse drinking approximately 6 mL of water per day. A stock solution was created of 1–2 mg of water-soluble DCZ into 1 mL of DMSO and vortexed until in solution. 17 µL of stock solution was combined with 1 L of diH$_2$O for a final concentration of 3 µg/kg/day per mouse, as previously described in *Suthard et al., 2023*, to avoid any potential excitotoxicity resulting through stimulating cells for up to 3 months, and to mimic a more sustained dysregulation that may accumulate in severity over time. Importantly, no obvious signs of distress or seizures were observed with chronic administration of DCZ by experimenters or vivarium staff for the entire duration of the project.

## Stereotaxic surgeries

For all surgeries, mice were initially anesthetized with 3.0% isoflurane inhalation during induction and maintained at 1–2% isoflurane inhalation through stereotaxic nose-cone delivery (oxygen 1 L/min). Ophthalmic ointment was applied to the eyes to provide adequate lubrication and prevent corneal desiccation. The hair on the scalp above the surgical site was removed using Veet hair removal cream and subsequently cleaned with alternating applications of betadine solution and 70% ethanol. 2.0% lidocaine hydrochloride (HCl) was injected subcutaneously as local analgesia prior to midsagittal incision of the scalp skin to expose the skull. 0.1 mg/kg (5 mg/kg) subcutaneous (SQ) dose of meloxicam was administered at the beginning of surgery. All animals received bilateral craniotomies with a 0.5–0.6 mm drill-bit for vCA1 injections. A 10 µL airtight Hamilton syringe with an attached 33-gauge beveled needle was slowly lowered to the coordinates of vCA1: –3.16 anteroposterior (AP), ± 3.10 mediolateral (ML) and both −4.25/−4.50 dorsoventral (DV) to cover vertical the axis of vHPC. All coordinates are given relative to bregma (mm). A total volume of 300 nL of AAV9-DIO-Flex-hM3Dq-mCherry or AAV9-DIO-Flex-mCherry was bilaterally injected into the vCA1 at both DV coordinates (150 nL at each site) listed above. The needle remained at the target site for 7–10 min post-injection before removal. Incisions were sutured closed using 4/0 non-absorbable Nylon Monofilament Suture [Brosan]. Following surgery, mice were injected with 0.1 mg/kg IP dose of buprenorphine (volume administered was dependent on the weight of the animal at the time of surgery). They were placed in a recovery cage with a heating pad until fully recovered from anesthesia. Histological assessment verified bilateral viral targeting and data from off-target injections were not included in analyses.

## Immunohistochemistry

Mice were overdosed with 3% isoflurane and perfused transcardially with cold (4°C) phosphate-buffered saline (PBS) followed by 4% paraformaldehyde (PFA) in PBS. Brains were extracted and kept in PFA at 4°C for 24–48 hr and transferred to a 30% sucrose in PBS solution. Long-term storage of the brains consisted of transferring to a 0.01% sodium azide in PBS solution until slicing. Brains were sectioned into 50-µm-thick coronal and sagittal sections with a vibratome and collected in cold PBS or 0.01% sodium azide in PBS for long-term storage. Sections underwent three washes for 10 min each in PBS to remove the 0.01% sodium azide that they were stored in. Sections were blocked for 2 hr at room temperature (RT) in PBS combined with 0.2% Triton X-100 (PBST) and 5% normal goat serum (NGS) on a shaker. Sections were incubated in the primary antibody (1:500 polyclonal guinea pig anti-NeuN/Fox3 [SySy #266-004]; 1:1000 mouse anti-GFAP [NeuroMab #N206A/8]; 1:1000 rabbit polyclonal anti-Iba1 [Wako #019-19741]) diluted in the same PBST-NGS solution at 4°C for 24–48 hr. Sections then underwent three washes for 10 min each in PBST. Sections were incubated in a secondary antibody (1:200 Alexa Fluor 488 anti-guinea [Invitrogen #A-11073], 1:1000 Alexa Fluor 488 anti-rabbit [Invitrogen #A-11008], 1:1000 Alexa Fluor 555 anti-mouse [Invitrogen #A-11001]) for 2 hr at RT. Sections then underwent three more washes in PBST for 10 min each. Sections were then mounted onto microscope slides (VMR International LLC, PA). Vectashield HardSet Mounting Medium with DAPI (Vector Laboratories Inc, CA) was applied and slides were coverslipped and allowed to dry for 24 hr at RT. Once dry, slides were sealed with clear nail polish around the edges and stored in a slide box in 4°C. If not mounted immediately, sections were stored in PBS at 4°C.

## Cell quantification and morphological analysis (NeuN, Iba1, GFAP)

Only animals that had accurate bilateral injections were selected for quantification. Images were acquired using a confocal microscope (Zeiss LSM800, Germany) with a ×20 objective. 6–8 single-tile images were taken in the brain regions of interest across 3–5 coronal slices, giving us a total of 18 single-tile images per mouse × 3–4 mice in each group. When imaging NeuN z-stacks, intervals were taken at 4 µm, whereas Iba1 and GFAP z-stacks were taken at 1 µm intervals for detailed morphological analysis. Ilastik (*Berg et al., 2019*) was used to quantify the total number of cells and 3DMorph (*York et al., 2018*) was used to assess the morphology of GFAP and Iba1 cells. Images of NeuN, GFAP, and Iba-1 were captured in a 300 µ 300 single-tile (micrometers) that contained matched ROIs across groups for hippocampal subregions.

For 3DMorph analysis, we selected a total of three single-tile images from the same ROI per mouse per group. This gave a total of ~20–30 cells per image. Therefore, we conducted morphological analysis on a total of 755 Iba1+ cells and 842 GFAP+ cells across all groups. Acquired image files (.czi)

were opened in ImageJ. Processing of images in *Figures 1, 2 and 5* and *Figure 1—figure supplement 1* involved maximizing intensity, removing outlier noise (despeckling), and adjusting contrast of images for improved detection of glial cells in 3DMorph.

Cell volume ($\mu m^3$) is calculated by converting the number of voxels occupied by the cell object to a real-world unit based on defined scaling factors. Territorial volume ($\mu m^3$) is estimated by creating a polygon around the outside points of each cell and measuring the volume inside. Ramification index is a measure of the complexity of astrocyte and microglial morphology, with a higher ramification index indicating a more complex and extensive branching structure. This is computed by dividing the cell territorial volume by the cell volume. Average centroid distance ($\mu m$) helps us understand the distribution of cells in the brain, with a larger centroid distance indicating the glial cells are further apart. Using a 3D reconstruction of each cell skeleton, branches are labeled as primary, secondary, tertiary, and quaternary. The number of branch points (#) is calculated by the intersections of each branch type and endpoints (#) are counted at the termination of these branches. Finally, each of these branches are measured in length and utilized for the minimum, maximum, and average branch length ($\mu m$).

## Behavioral assays

All behavioral assays were conducted during the light cycle of the day (07:00–19:00) on animals. Mice were handled for 3–5 days, 2 min per day, before all behavioral experiments began. Behavioral assays include open field, zero maze, y-maze, fear conditioning, recall, extinction, and generalization. As described in our previous work in greater detail (*Suthard et al., 2023*), DCZ water was removed 1–2 hr from the home cage to allow for a decline in ligand concentration in the brain and to ensure there were no acute effects of the drug on behavior. When administered IP in mice, DCZ peaks 5–10 min after injection, declining to baseline levels around 2 hr after (*Nagai et al., 2020*).

## Open-field test

An open (61 cm × 61 cm) arena with black plastic walls was used for the open-field test, with a red-taped area in the middle delineating 'center' from 'edges'. A camera was placed above the open field in order to record video of the session. Mice were individually placed into the center of the chamber and allowed to explore freely for 10 min. At the end of the session, each mouse was placed into a separate cage until all of its cage mates had also gone through the behavioral test. They were all placed back into the home cage once this occurred. An automated video-tracking system, AnyMaze v4.3 (Stoelting Co, Wood Dale, IL), was used to measure total distance traveled, number of entries into the center, and total time spent in the center.

## Zero-maze test

The zero-maze test is a pharmacologically validated assay of anxiety in animal models that is based on the natural aversion of mice to elevated, open areas. It is composed of a 44-cm-wide ring with an outer diameter of 62 cm. It contained four equal zones of walled (closed) and unwalled (open) areas. The entire ring is 8 cm in height and sits at an elevation of 66 cm off the ground. All animals at each time point were tested on the same day. Mice were placed in the closed area at the start of a 10 min session. The following parameters were analyzed: time spent in the open area, number of entries into the open area, etc. The maze was cleaned with 70% ethanol between mice. During the behavioral testing, the lighting levels remained constant and there were no shadows over the surface of the maze.

## y-maze test

The y-maze is a hippocampal-dependent spatial working memory task that requires mice to use external cues to navigate the identical internal arms. The apparatus consisted of a clear plexiglass maze with three arms (36.5 cm length, 7.5 cm width, 12.5 cm height) that were intersected at 120°. A mouse was placed at the end of one arm and allowed to move freely through the maze for 10 min without any reinforcements, such as food or water. Entries into all arms were noted (center of the body must cross into the arm for a valid entry) and a spontaneous alternation was counted if an animal entered three different arms consecutively. Percentage of spontaneous alternation was calculated according to the following formula: [(number of alterations)/(total number of arm entries - 1) × 100]. To prevent bias in data analysis, the test was carried out in a blind manner by the experimenter and

behavior was analyzed blindly in AnyMaze. Additionally, total distance traveled was calculated for the test to rule out changes in locomotion.

### Fear conditioning, recall, and extinction

For CFC, all time points took place in mouse conditioning chambers (Coulbourn Instruments, Holliston, MA). The grid floor was connected to a precision animal shocker (Coulbourn Instruments), which delivered a total of four foot shocks (2 s duration, 1.5 mA) at the 120, 180, 240, and 300 s time points during the 360 s session. This mouse conditioning chamber (18.5 × 18.5 × 21.5 cm) was composed of metal walls, plexiglass front and back walls, and a stainless-steel grid floor (16 grid bars). For recall, all animals were placed back into the same chambers they were originally fear conditioned in for a total of 300 s in the absence of foot shocks. For extinction, all animals were placed back into the same chambers they were originally conditioned in for a total of 1800s in the absence of foot shocks. To record the animal's freezing activity, a video camera was mounted to the ceiling of the chamber and fed into a computer running FreezeFrame software (Actimetrics, Wilmette, IL). The program determined movement as changes in pixel luminance over a set period of time. Freezing was defined as a bout of 1.25 s or longer without changes in pixel luminance and verified by an experimenter blind to treatment groups. The chambers were cleaned with 70% ethanol solution prior to each animal placement.

### Context B exposure

For generalization (context B), after 5 days of extinction, the animals were taken into a different room from the original conditioning and placed in a new larger chamber. During the experiment, the floor was changed to a smooth bottom, the walls were changed to have striped backgrounds, the odor was changed to a neutral almond scent, and the lights were turned off during the procedure. Red lights were turned on in the room for the experimenter. The entire exposure to context B was a total of 10 min. Activity recording and freezing level analysis were assessed as described above using FreezeFrame software. The chambers were cleaned with 70% ethanol solution prior to each animal placement.

### Statistics and data analysis

Data were analyzed using GraphPad Prism (GraphPad Software v9.4.1, La Jolla, CA). Data were tested for normality using the Shapiro–Wilk and Kolmogorov–Smirnov tests. Outliers were removed using the ROUT method, which uses identification from nonlinear regression. We chose a ROUT coefficient Q value of 10% (false discovery rate), making the threshold for outliers less-strict and allowing for an increase in power for outlier detection. No more than 1–2 mice were removed in any group as outliers for any behavioral measure. To analyze differences between groups (mCherry and hM3Dq) and across age groups (3 and 11 months old [fear conditioning]; 6 and 14 months old [all other behavioral tasks]), we used a two-way ANOVA (between-subject factor: group; within-subject factor: age). To analyze the differences between groups and across time within a single session (i.e., fear conditioning) or across months (i.e., body weight), we used two-way RM ANOVAs (between-subjects factor: group; within-subjects factor: time/month). Post hoc analyses were performed using Tukey's and Sidak's multiple comparisons test, where applicable. Alpha was set to 0.05 for all tests.

## Acknowledgements

This material was based on the work supported by an NIH Early Independence Award DP5 OD023106-01, NIH Transformative Award, the Ludwig Family Foundation, the Air Force Office of Scientific Research award (FA9550-21-1-0310), the Pew Scholars Program in the Biomedical Sciences, the Chan-Zuckerberg Initiative, and the Center for Systems Neuroscience and Neurophotonics Center at Boston University.

# Additional information

## Funding

| Funder | Grant reference number | Author |
|---|---|---|
| National Institutes of Health | Transformative Award | Steve Ramirez |
| Ludwig Family Foundation | | Rebecca L Suthard |
| Air Force Office of Scientific Research | FA9550-21-1-0310 | Steve Ramirez |
| Pew Scholars Program in Biomedical Sciences | | Steve Ramirez |
| Chan-Zuckerberg Initiative | | Steve Ramirez |
| Center for Systems Neuroscience and Neurophotonics Center at Boston University | | Steve Ramirez |
| Nationai Institutes of Health Early Independence Award | DP5 OD023106-01 | Steve Ramirez |

The funders had no role in study design, data collection and interpretation, or the decision to submit the work for publication.

## Author contributions

Alexandra L Jellinger, Conceptualization, Data curation, Formal analysis, Writing – original draft, Writing – review and editing; Rebecca L Suthard, Conceptualization, Data curation, Formal analysis, Methodology, Writing – original draft, Writing – review and editing; Bingbing Yuan, Data curation, Methodology, Writing – original draft, Writing – review and editing; Michelle Surets, Data curation, Formal analysis, Writing – original draft, Writing – review and editing; Evan A Ruesch, Data curation, Formal analysis, Validation, Writing – original draft, Writing – review and editing; Albit J Caban, Data curation, Methodology; Shawn Liu, Conceptualization, Data curation, Formal analysis, Supervision, Methodology, Writing – original draft, Project administration, Writing – review and editing; Monika Shpokayte, Data curation, Formal analysis, Methodology, Writing – original draft, Writing – review and editing; Steve Ramirez, Conceptualization, Supervision, Funding acquisition, Investigation, Methodology, Writing – original draft, Project administration, Writing – review and editing

## Author ORCIDs

Rebecca L Suthard ⓘ https://orcid.org/0000-0002-0011-7028
Shawn Liu ⓘ https://orcid.org/0000-0002-2799-2519
Steve Ramirez ⓘ http://orcid.org/0000-0002-9966-598X

## Ethics

All subjects were treated in accord with protocol 201800579 approved by the Institutional Animal Care and Use Committee (IACUC) at Boston University.

Reviewer #1 (Public Review): https://doi.org/10.7554/eLife.96281.3.sa1
Reviewer #2 (Public Review): https://doi.org/10.7554/eLife.96281.3.sa2
Reviewer #3 (Public Review): https://doi.org/10.7554/eLife.96281.3.sa3
Author response https://doi.org/10.7554/eLife.96281.3.sa4

# Additional files

## Supplementary files

• MDAR checklist

## Data availability

Raw data along with gene expression levels are deposited to NCBI Gene Expression Omnibus under accession code GSE198731.

The following dataset was generated:

| Author(s) | Year | Dataset title | Dataset URL | Database and Identifier |
|---|---|---|---|---|
| Liu S, Yuan B | 2022 | Hippocampal cells multiplex positive and negative engrams | https://www.ncbi.nlm.nih.gov/geo/query/acc.cgi?acc=GSE198731 | NCBI Gene Expression Omnibus, GSE198731 |

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
