## [Editor Report · eLife assessment]

This **useful** study reports the behavioral and physiological effects of the longitudinal activation of neurons associated with negative experiences. The main claims of the article are supported by **solid** experimental evidence, although the specificity of the long-term manipulation could have benefitted from additional validation. This study will be of interest to neuroscientists working on memory.

---

## [Referee Report · Reviewer #1 (Public Review)]

Summary:

In this study, Jellinger et al. performed engram-specific sequencing and identified genes that were selectively regulated in positive/negative engram populations. In addition, they performed chronic activation of the negative engram population over 3 months and observed several effects on fear/anxiety behavior and cellular events such as upregulation of glial cells and decreased GABA levels.

Strengths:

They provide useful engram-specific GSEA data and the main concept of the study, linking negative valence/memory encoding to cellular level outcomes including upregulation of glial cells, is interesting and valuable.

Comments on the revised manuscript:

The revised manuscript still does not adequately address the primary technical concern regarding long-term DREADD manipulation. The authors reference their previous work (Suthard et al., 2023) as evidence; however, this earlier paper only presents fluorescence intensity in a non-quantitative manner with merely three samples (Supplementary Figure 7). This limited evidence does not sufficiently support the claim of potent long-term activation. The discussion in the revision stating "...even if our manipulation is only working for 1 month, rather than 3 months..." is unconvincing, particularly given that the title and abstract still claims "chronic activation of...". To substantiate the technical validity of the study, at least cFos staining at various time points is necessary, which is less burdensome compared to more direct demonstrations such as slice physiology. Thus, although I believe it could be an interesting study for some audiences, I cannot support the strength of the evidence presented in the study.

Furthermore, in response to all reviewers' concerns regarding the quantification of GABA, the authors have removed the data from the study rather than providing properly acquired images or quantified data. This action diminishes the significance of the study.

---

## [Referee Report · Reviewer #2 (Public Review)]

Summary:

Jellinger, Suthard, et al. investigated the transcriptome of positive and negative valence engram cells in the ventral hippocampus, revealing anti- and pro-inflammatory signatures of these respective valences. The authors further reactivated the negative valence engram ensembles to assay the effects of chronic negative memory reactivation in young and old mice. This chronic re-activation resulted in differences in aspects of working memory, fear memory, and caused morphological changes in glia. Such reactivation-associated changes are putatively linked to GABA changes and behavioral rumination.

Strengths:

Much the content of of this manuscript is of benefit to the community, such as the discovery of differential engram transcriptomes dependent on memory valence. The chronic activation of neurons, and the resultant effects on glial cells and behavior, also provide the community with important data. Laudable points of this manuscript include the comprehensiveness of behavioral experiments, as well as the cross-disciplinary approach.

Weaknesses:

Weaknesses noted in the previous version of the manuscript have been accounted for.

---

## [Referee Report · Reviewer #3 (Public Review)]

Summary:

The authors note that negative ruminations can lead to pathological brain states and mood/anxiety dysregulation. They test this idea by using mouse engram-tagging technology to label dentate gyrus ensembles activated during a negative experience (fear conditioning). They show that chronic chemogenetic activation of these ensembles leads to behavioral (increased anxiety, increased fear generalization, reduced fear extinction) and neural (increases in neuroinflammation, microglia and astrocytes).

Strengths:

The question the authors ask here is an intriguing one, and the engram activation approach is a powerful way to address the question. Examination of a wide range of neural and behavioral dependent measures is also a strength.

Weaknesses:

The major weakness is that the authors have found a range of changes that are correlates of chronic negative engram reactivation. However, they do not manipulate these outcomes to test whether microglia, astrocytes, neuroinflammation are causally linked to the dysregulated behaviors.

---

## [Author Response]

The following is the authors’ response to the original reviews.

**Public Reviews:**

**Reviewer #1 (Public Review):**
Summary:In this study, Jellinger et al. performed engram-specific sequencing and identified genes that were selectively regulated in positive/negative engram populations. In addition, they performed chronic activation of the negative engram population over 3 months and observed several effects on fear/anxiety behavior and cellular events such as upregulation of glial cells and decreased GABA levels.Strengths:They provide useful engram-specific GSEA data and the main concept of the study, linking negative valence/memory encoding to cellular level outcomes including upregulation of glial cells, is interesting and valuable.Weaknesses:A number of experimental shortcomings make the conclusion of the study largely unsupported. In addition, the observed differences in behavioral experiments are rather small, inconsistent, and the interpretation of the differences is not compelling.Major points for improvement:(1) Lack of essential control experimentsWith the current set of experiments, it is not certain that the DREADD system they used was potent and stable throughout the 3 months of manipulations. Basic confirmatory experiments (e.g., slice physiology at 1m vs. 3m) to show that the DREADD effects on these vHP are stable would be an essential bottom line to make these manipulation experiments convincing.

In previous work from our lab performing long-term activation of Gq DREADD receptors in the vHPC, we quantify the presence of Gq receptor expression over 3-, 6- and 9-month timepoints and show that there is no decrease in receptor expression, as measured via fluorescence intensity (Suthard et al., 2023). In this study, we also address that even if our manipulation is only working for 1 month, rather than 3 months, we are observing the long-term effects of this shorter-term stimulation. This is still relevant, and only changes how we interpret these findings, as shorter-term stimulation or disruption of neuronal activity can still have detrimental effects on behavior.

Furthermore, although the authors use the mCherry vector as a control, they did not have a vehicle/saline control for the hM3Dq AAV. Thus, the long-term effects such as the increase in glial cells could simply be due to the toxicity of DREADD expression, rather than an induced activity of these cells.

For chemogenetic studies, our experimental rationale utilized a standard approach in the field, which includes one of two control options: (1) active receptor vs. control vector + ligand or (2) active receptor + ligand or saline control. We chose the first option, as this more properly controls for the potential off-target effects of the ligand itself, as shown in other previous work (Xia et al., 2017). This is particularly important for studies using CNO, as many off-target effects have been noted as a limitation (Manvich et al., 2018). We chose to use DCZ as it is closely related to CNO and newer ligands, but comes with added benefits of high specificity, low off-target effects, high potency and brain penetrance (Nagai et al., 2020), but any potential off-target effects of DCZ are yet to be completely investigated as this ligand is very new.

Evidence of DREADD toxicity has been shown at high titer levels of AAV2/7- CamKIIα-hM4D(Gi)-mCherry in the hippocampus at 5 weeks, as the reviewer pointed out in their above comment (Goossens et al., 2021). Our viral strategy is targeted to a much smaller number of cells using AAV9-DIO-Flex-hM3Dq-mCherry at a lower titer, unlike expression within a much larger population of CaMKII+ excitatory neurons in this study. Additionally, visual comparison of their viral load and expression with ours shows much more intense expression that spans a larger area of the hippocampus (Goossens et al, 2021; Figure 1D), whereas ours is isolated to a smaller region of vHPC (see Figure 1B).

Further, we attempted to quantify a decrease in neuronal health (Yousef et al., 2017) resulting from DREADD expression via NeuN counts within multiple hippocampal subregions for the 6- and 14-month groups across active Gq receptor and mCherry conditions and did not observe significant decreases in NeuN as a result (Supplemental Figure 1). However, immunohistochemistry of an individual marker may not be sufficient to capture the entire health profile of an individual neuron and future work should consider other markers of cell death or inflammation, which we have added to the Limitations & Future Work section of our Discussion.

(2) Figure 1 and the rest of the study are disconnectedThe authors used the cFos-tTA system to label positive/negative engram populations, while the TRAP2 system was used for the chronic activation experiments. Although both genetic tools are based on the same IEG Fos, the sensitivity of the tools needs to be validated. In particular, the sensitivity of the TRAP2 system can be arbitrarily altered by the amount of tamoxifen (or 4OHT) and the administration protocols. The authors should at least compare and show the percentage of labeled cells in both methods and discuss that the two experiments target (at least slightly) different populations. In addition, the use of TRAP2 for vHP is relatively new; the authors should confirm that this method actually captures negative engram populations by checking for reactivation of these cells during recall by overlap analysis of Fos staining or by artificial activation.

We thank the reviewer for their comments and opportunity to discuss the marked differences between TRAP2 and DOX systems. In particular, we agree that while both systems rely on the the Fos promoter to drive an effector of interest, their efficacy and temporal resolution vary substantially depending on genetic cell-type, brain region, temporal parameters of Dox or 4-OHT delivery, subject-by-subject metabolic variability, and threshold to Fos induction given the promoter sequences inherent to each system. For example, recent studies have reported the following:

- The TRAP2 line labels a subset of endogenously activeCA1 pyramidal cells (e.g. 5-18%) while the DOX system labels 20-40% of CA1 pyramidal cells (DeNardo et al, 2019; Monasterio et al, *BioRxiv* 2024 ).

- The temporal windows for each range from hours in TRAP2 to 24-48 hours for DOX (DeNardo et al, 2019; Denny et al, 2014; Liu & Ramirez et al, 2012).

- The efficacy of “tagging” a population of cells with TRAP2 vs. with DOX will constrain the number of possible cells that may overlap with cFos upon re-exposure to a given experience e.g. see the observed overlaps in vCA1 - BLA circuits (Kim & Cho, 2020), compared to vCA1 in general (Ortega-de San Luis et al, 2023) and valence-specific vCA1 populations (Shpokayte et al, 2022).

- Tagging vCA1 cells with both the TRAP2 and DOX systems are nonetheless sufficient to drive corresponding behaviors (e.g. vCA1 terminal stimulation drives behavioral changes with the DOX and TRAP2 system (Shpokayte et al, 2022) and vCA1 stimulation of an updated fear-linked ensemble drives light-induced freezing in a neutral context utilizing the TRAP2 and DOX systems (Ortega-de San Luis et al, 2023)).

Finally, and promisingly, as more studies continue to link the in vivo physiological dynamics of these cell populations tagged using each system (e.g. compare Pettit et al, 2022 with Tanaka et al, 2018) and correlating their activity to behavioral phenotypes, our field is in the prime position to uncover deeper principles governing hippocampus-mediated engrams in the brain. Together, we believe a more comprehensive understanding of these systems is fully warranted, especially in the service of further cataloging cellular similarities and differences within such tagged populations.

(3) Interpretation of the behavior dataIn Figures 3a and b, the authors show that the experimental group showed higher anxiety based on time spent in the center/open area. However, there were no differences in distance traveled and center entries, which are often reduced in highly anxious mice. Thus, it is not clear what the exact effect of the manipulation is. The authors may want to visualize the trajectories of the mice's locomotion instead of just showing bar graphs.

Our findings show that our experimental group displays higher levels of anxiety-like behaviors as measured via time spent in center/open area, while there are no differences in distance traveled or center entries. For distance traveled, our interpretation is in line with complementary research (Jimenez et al, 2018; Kheirbek et al, 2013) that shows no changes in distance traveled/distance traveled in the center coupled with changes in anxiety levels as a result of manipulation within anxiety-related circuits. More broadly, any locomotion-related deficit could cause a change in distance traveled that is unrelated to anxiety-like behaviors alone. For example, a reduction in distance traveled could be coupled with a decrease in time spent in the center, but could also result only from motor or exploratory deficits. We hope that this explanation clarifies our interpretation of the open field and elevated plus maze findings in light of other literature.

In addition, the data shown in Figure 4b is somewhat surprising - the 14MO control showed more freezing than the 6MO control, which can be interpreted as "better memory in old". As this is highly counterintuitive, the authors may want to discuss this point. The authors stated that "Mice typically display increased freezing behavior as they age, so these effects during remote recall are expected" without any reference. This is nonsense, as just above in Figure 4a, older mice actually show less freezing than young mice. Overall, the behavioral effects are rather small and random. I would suggest that these data be interpreted more carefully.

In Figure 4B, we present our findings from remote recall and observe increased freezing levels in control mice with age, as mentioned by the reviewer, indicating increased memory. This is in line with previous work from Shoji & Miyakawa, 2019 which has been added as a reference for the quotation described above; we thank the reviewer for pointing this error out. As the reviewer has pointed out, above in Figure 4A, we measured freezing levels across all groups during contextual fear conditioning before the start of chronic stimulation, as this was the session we ‘tagged’ a negative memory in. Although it appears that there may be slightly lower levels of freezing in older (14-month old) mice, our findings do not determine statistical significance for difference between age group, only effects of time and subject which are expected as freezing increases within the session and animals display high levels of variability in freezing levels across many experiments (Figure 4A i-iii). We also find in previous work that control mice receiving 3-, 6- and 9-months of chronic DCZ stimulation in the vHPC with empty vector (mCherry) receptor show an increase in freezing with age (Suthard et al, 2023; Figure 2A ii).

(4) Lack of citation and discussion of relevant studyKhalaf et al. 2018 from Gräff lab showed that experimental activation of recall-induced populations leads to fear attenuation. Despite the differences in experimental details, the conceptual discrepancy should be discussed.

As mentioned by the reviewer, Khalaf et al. 2018 showed that experimental activation of recall-induced populations in the dentate gyrus leads to fear attenuation. Specifically, they pose that this fear attenuation occurs in these ensembles through updating or unlearning of the original memory trace via the engagement, rather than suppression, of an original traumatic experience. Despite the differences in experimental details with our current study and this work, we agree that the conceptual discrepancy should be discussed. First, one major difference is that we are reactivating an ensemble that was tagged during fear memory encoding, while Khalaf et al. are activating a remote recall-induced ensemble that was tagged one month after encoding. Although there is high overlap between the encoding and recall ensembles when mice are exposed to the conditioning context, these ensembles are not identical and may result in different behavioral phenotypes when chronically reactivated. Further, Khalaf et al rely on reactivation of the recall-induced ensemble during extinction to facilitate rapid fear attenuation. This differs from our current work, as their reactivation is occurring during the extinction process in the previously conditioned context, while we are reactivating chronically in the animal’s home cage over the course of a longer time period. It may be necessary that the memory is first reactivated, and thus, more liable to re-contextualization, in the original context compared to an unrelated homecage environment where there are presumably no related cues present. Importantly, this previous work tests the attenuation of fear shortly after an extinction process, while we are not traditionally extinguishing the context with aid of the memory reactivation. Finally, we are testing remote recall (3 months post-conditioning), while they are testing at a shorter time interval (28 days). In line with these ideas, future work may seek to tease out the mechanistic differences between recent and remote memory extinction both in terms of natural memory recall and chronically manipulated memory-bearing cells.

**Reviewer #2 (Public Review):**
Summary:Jellinger, Suthard, et al. investigated the transcriptome of positive and negative valence engram cells in the ventral hippocampus, revealing anti- and pro-inflammatory signatures of these respective valences. The authors further reactivated the negative valence engram ensembles to assay the effects of chronic negative memory reactivation in young and old mice. This chronic re-activation resulted in differences in aspects of working memory, and fear memory, and caused morphological changes in glia. Such reactivation-associated changes are putatively linked to GABA changes and behavioral rumination.Strengths:Much of the content of this manuscript is of benefit to the community, such as the discovery of differential engram transcriptomes dependent on memory valence. The chronic activation of neurons, and the resultant effects on glial cells and behavior, also provide the community with important data. Laudable points of this manuscript include the comprehensiveness of behavioral experiments, as well as the cross-disciplinary approach.Weaknesses:There are several key claims made that are unsubstantiated by the data, particularly regarding the anthropomorphic framing of "rumination" on a mouse model and the role of GABA. The conclusions and inferences in these areas need to be carefully considered.(1) There are many issues regarding the arguments for the behavioural data's human translation as "rumination." There is no definition of rumination provided in the manuscript, nor how rumination is similar/different to intrusive thoughts (which are psychologically distinct but used relatively interchangeably in the manuscript), nor how rumination could be modelled in the rodent. The authors mention that they are attempting to model rumination behaviours by chronically reactivating the negative engram ("To understand if our experimental model of negative rumination..."), but this occurs almost at the very end of the results section, and no concrete evidence from the literature is provided to attempt to link the behavioural results (decreased working memory, increased fear extinction times) to rumination-like behaviours. The arguments in the final paragraph of the Discussion section about human rumination appear to be unrelated to the data presented in the manuscript and contain some uncited statements. Finally, the rumination claims seem to be based largely upon a single data figure that needs to be further developed (Figure 6, see also point 2 below).(2) The staining and analysis in Figure 6 are challenging to interpret, and require more evidence to substantiate the conclusions of these results. The histological images are zoomed out, and at this resolution, it appears that only the pyramidal cell layer is being stained. A GABA stain should also label the many sparsely spaced inhibitory interneurons existing across all hippocampal layers, yet this is not apparent here. Moreover, both example images in the treatment group appear to have lower overall fluorescence intensity in both DAPI and GABA. The analysis is also unclear: the authors mention "ROIs" used to measure normalized fluorescence intensity but do not specify what the ROI encapsulates. Presumably, the authors have segmented each DAPI-positive cell body and assessed fluorescence however, this is not explicated nor demonstrated, making the results difficult to interpret.

Based on the collective discussion from all reviewers on the completeness of our GABA quantification and its implications, we have decided to remove this figure and perform more substantive analysis of this E/I imbalance in future work.

(3) A smaller point, but more specific detail is needed for how genes were selected for GSEA analysis. As GSEA relies on genes to be specified a priori, to avoid a circular analysis, these genes need to be selected in a blind/unbiased manner to avoid biasing downstream results and conclusions. It's likely the authors have done this, but explicitly noting how genes were selected is an important context for this analysis.

As mentioned in our Methods section, gene sets were selected based on pre-existing biology and understanding of genes canonically involved in “neurodegeneration” such as those related to apoptotic pathways and neuroinflammation or “neuroprotection” such as brain-derived neurotrophic factor, to name a few. A limitation of this method is that we must avoid making strong claims about the actual function of these up- or down-regulated genes without performing proper knock-in or knock-out studies, but we hope that this provides an unbiased inventory for future experiments to perform causal manipulations.

**Reviewer #3 (Public Review):**
Summary:The authors note that negative ruminations can lead to pathological brain states and mood/anxiety dysregulation. They test this idea by using mouse engram-tagging technology to label dentate gyrus ensembles activated during a negative experience (fear conditioning). They show that chronic chemogenetic activation of these ensembles leads to behavioral (increased anxiety, increased fear generalization, reduced fear extinction) and neural (increases in neuroinflammation, microglia, and astrocytes).Strengths:The question the authors ask here is an intriguing one, and the engram activation approach is a powerful way to address the question. Examination of a wide range of neural and behavioral dependent measures is also a strength.Weaknesses:The major weakness is that the authors have found a range of changes that are correlates of chronic negative engram reactivation. However, they do not manipulate these outcomes to test whether microglia, astrocytes, or neuroinflammation are causally linked to the dysregulated behaviors.
**Recommendations For The Authors:**

**Reviewer #1 (Recommendations For The Authors):**
- Figure 2c should include Month0, the BW before the start of the manipulation.

Regrettably, we do not have access to the Month 0 body weights at this time as this project changed hands over the course of the past year or so. This is an inherent limitation that we missed during analysis and we pose this as a limitation in the Results section after describing this finding. Therefore, it is possible that over the first month of stimulation (Month 0-1), there may have been a drop in body weight that rebounded by the first measurement at Month 1 that continued to increase normally through Months 2-3, as shown in our Figure 1. Thank you for this note.

- Figure 6a looks confusing - the background signal in the green channel is very different between control and experimental groups. Were representative images taken with different microscope settings?

The representative images were taken with the same microscope power settings, but were adjusted in brightness/contrast within FIJI for clarity in the Figure – we apologize that this was misleading in any way and thank the reviewer for their feedback. Further, based on the collective discussion from all reviewers on the completeness of our GABA quantification and its implications, we have decided to remove this figure and perform more substantive analysis of this E/I imbalance in future work.

- Typo mChe;try

This typo was fixed

- "During this contextual... mice in the 6- and 14- month groups..." Isn't it 3- and 11- month respectively at the time of fear conditioning? Throughout the manuscript, this point was written very confusingly.

Yes, we thank the reviewer for pointing this out. It has been corrected to 3- and 11-month old mice at the timing of fear conditioning and clarified throughout the manuscript where applicable.

- "GABAergic eYFP fluorescence" Where does the eYFP come from? The methods state that GABA quantification is based on IHC staining.

Based on the collective discussion from all reviewers on the completeness of our GABA quantification and its implications, we have decided to remove this figure and perform more substantive analysis of this

E/I imbalance in future work. We discuss this E/I balance not being directly assessed in the Limitations & Future Directions section of our Discussion, noting the importance of detailed quantification of both excitatory and inhibitory markers within the hippocampus.

**Reviewer #2 (Recommendations For The Authors):**
(1) There is a full methods section ("Analysis of RNA-seq data") that mostly describes RNA-seq analysis that seemingly does not appear in the paper. This section should be reviewed.

We have included this portion of the methods that explain the previous workflow from Shpokayte et al., 2022 where this dataset was generated and this has been noted in the “Analysis of RNA-seq data” section of the methods.

(2) Figure 6: GABA staining should be more critically analyzed, as discussed above, and validated with another GABA antibody for rigor. From the representative images provided in Figure 6, it looks possibly as though the hM3Dq images were simply not fully in the focal plane when being imaged or were over-washed, as DAPI staining also appears to be lower in these images.

Based on the collective discussion from all reviewers on the completeness of our GABA quantification and its implications, we have decided to remove this figure and perform more substantive analysis of this E/I imbalance in future work. Specifically, it will be necessary to rigorously investigate both excitatory and inhibitory markers within this region to ensure these claims are substantiated. Thank you for this suggestion.

(3) The first claim that human GABAergic interneurons cause rumination is uncited. (Page 19, first sentence beginning with: "Evidence from human studies suggests...").

Based on the collective discussion from all reviewers on the completeness of our GABA quantification and its implications, we have decided to remove this figure and perform more substantive analysis of this E/I imbalance in future work. Apologies for the lack of citation in-text, the proper citation for this finding is Schmitz et al, 2017.

(4) Gene names throughout the manuscript and figure are written in the wrong format for mice (eg: Page 13, second line: SPP1, TTR, and C1QB1 instead of Spp1, Ttr, C1qb1).

This was corrected throughout the manuscript.

(5) Tense on Page 15 third sentence of the second paragraph: "...spatial working memory was assessed...".

This was corrected throughout the manuscript.

(6) Supplemental Figure 1 would benefit from normalization of the NeuN+ cell counts. The inclusion of an excitatory and inhibitory neuron marker in this figure might benefit the argument that there is a change in the excitation/inhibition of the hippocampus - as the numbers of excitatory neurons outweigh the numbers of inhibitory neurons that would be assayed here.

In an effort to normalize the NeuN+ cell counts, for each of our ROIs (6-8 single tiles for each brain region (DG, vCA1, vSub) x 3-5 coronal slices = ~18 single tiles per mouse x 3-4 mice) we captured a 300 x 300 micrometer, single-tile z-stack at 20x magnification. These ROIs were matched for dimensions and brain regions across all groups for each hippocampal subregion quantified. We initially proposed to normalize these NeuN counts over DAPI, but because DAPI includes all nuclei (microglia, oligodendrocytes, astrocytes and neurons), we weren’t sure this was the most optimal tool. We do agree that further quantification of excitatory and inhibitory cell markers would be vital to more concrete interpretation of our findings and we have added this to our Limitations & Future Work section of the Discussion.

**Reviewer #3 (Recommendations For The Authors):**
(1) The DOX tagging window lacks temporal precision. I suggest the authors note this as a limitation.

We thank the reviewer for noting this, and we have added this limitation to the Methods section with the context of the 24-48 hour DOX window being longer than other methods like TRAP.

(2) Is there a homeostatic response to chronic engram stimulation? That is, is DCZ as effective in increasing neuronal excitability on day 90 as it is on day 1. This could be addressed with electrophysiology, or with IEG induction. Alternatively, the authors could refer to previous literature-- for example, Xia et al (2017) eLife-- that examined whether there was any blunting of the effects of DREADD ligands after sustained delivery via drinking water. There, of course, may be other papers as well.

As noted by the reviewer, it is important to determine if DCZ maintains its effects on neuronal excitability throughout the 3 month administration period. To address this, previous work has shown that CNO administration in drinking water over one month consistently inhibited hM4Di+ neurons without altering baseline neuronal excitability as measured by firing rate and potassium currents (Xia et al, 2017). Although this is only for one month, it is administered via the same oral route as our DCZ protocol and suggests that at least for that amount of time we are likely producing consistent effects. In our reply above to Reviewer #1’s comment, we also note that even if DCZ is only having an effect for one month, rather than 3 months, we are still observing enduring changes that resulted from this short-term disturbance.

(3) Please double check there is no group effect on weight in 6-month-old mice in Figure 2C.

Two-way RM ANOVA showed no main effect of Group within the 6-month-old control and hM3Dq groups.

Group: F(1,17) = 1.361, p=0.2594.

(4) The shock intensity is much higher than is typical for fear conditioning studies in mice. Why was this the case?

Yes, we do agree that this shock intensity is on the higher side of typical paradigms in mice, however, our lab has utilized 0.75mA to 1.5mA intensity foot shocks for contextual fear conditioning in the past (Suthard & Senne et al, 2023; 2024; Dorst & Senne et al, 2023; Grella et al., 2022; Finkelstein et al., 2022) and we maintained this protocol for internal consistency. However, it would be interesting to systematically investigate how differing intensities of foot shock, subsequent tagging of this ensemble and reactivation would uniquely impact behavioral state acutely and chronically in mice.

(5) Remote freezing is very low. The authors should comment on this-- perhaps repeated testing has led to some extinction?

A reviewer above suggested a similar phenomenon may be occuring, specifically fear attenuation as a result of chronic stimulation. They referenced previous work from Khalaf et al. 2018, where they reactivated a recall-induced ensemble, while we reactivated an ensemble tagged during encoding. We expand upon this work in light of our findings within the Limitations & Future Work section of our Discussion. However, we do appreciate the lower levels of freezing observed in remote recall and sought out other literature to understand the typical range of remote freezing levels. One thing that we note is that our remote recall is occurring 3 months after conditioning, which is much longer than typical 14-28 day protocols. However, we find that freezing levels at remote timepoints from 21-45 days results in contextual freezing levels of between 20-50% approximately (Kol et al., 2020), as well as 40-75% approximately in a variety of 28 day remote recall experiments (Lee et al., 2023). This information, together with our current experimental protocol demonstrates a wide range of remote freezing levels that may depend heavily on the foot shock intensity, duration of days after conditioning, and animal variability.

(6) "mice display increased freezing with age": please add a reference.

Apologies, we missed the citation for that claim and it has been added in-text and in the references list (Shoji & Miyakawa, 2019).

(7) Related to the low freezing levels for remote memory, why is generalization minimal? Many studies have shown that there is a time-dependent emergence of generalized fear, yet here this is not seen. Is it linked to extinction (as above)? Or genetic background?

Previous work has shown that rats receiving multiple foot shocks during conditioning displayed a time-dependent generalization of context memory, while those receiving less shocks did not (Poulos et al., 2016), as the reviewer noted in their comment. In our current study, we observe low levels of generalization in all of our groups compared to freezing levels displayed in the conditioned context at the remote timepoint, in opposition to this time-dependent enhancement of generalization. It is possible that the genetic background of our C57BL/6J mice compared to the Long-Evans rat strain in this previous work accounts for some of this difference. In addition, it is possible that the longer duration of time (3 months) compared to their remote timepoint (28 days) resulted in time-dependent decrease in generalization that decreases with greater durations of time from original conditioning. As noted above, it is indeed plausible that the reactivation of a contextual fear ensemble over time is attenuating freezing levels for both the original and similar contexts (Khalaf et al, 2018). We discuss the differences in our study and this 2018 work more comprehensively above.

(8) Morphological phenotypes of astrocytes/microglia. Would be great to do some transcriptomic profiling of microglia/astrocytes to couple with the morphological characterization (but appreciate this is beyond the scope of current work).

We thank the reviewer this suggestion, we agree that would be an incredibly informative future experiment and have added this to our Limitations & Future Experiments section of the Discussion.

(9) The authors could consider including a limitations section in their discussion which discusses potential future directions for this work:- causal experiments.- E/I balance is not assessed directly (interestingly, in this regard, expanded engrams are linked to increased generalization [e.g., Ramsaran et al 2023]).

Thank you for this suggestion, we have added a Limitations & Future Directions section to our Discussion and have expanded upon these suggested points.

(10) For Figure 10, consider adding an experimental design/timeline.

We are making the assumption that the reviewer meant Figure 1 instead of Figure 10 here, but note that there is a description of the viral expression duration (D0-D10), followed by an off Dox period of 48 hours (D10-D12), with subsequent engram tagging of a negative (foot shock) or positive (male-to-female exposure) on D12. In our experiments (Shpokayte et al., 2022), Dox was administered for 24 hours (D12-D13), which was followed by sacrificing the animal for cell suspension and sequencing of the positive and negative engram populations. This figure also shows the viral strategy for the Tet-tag system (Figure 1A), as well as representative viral expression in vHPC (Figure 1B). We are happy to add additional experimental design/timeline information to this figure that would be helpful to the reviewer.